*Resource*

# Distinct senotypes in p16- and p21-positive cells across human and mouse aging tissues

Dominik Saul [1,2,3,4] ✉, Diana Jurk [2,5], Madison L Doolittle [1,2], Robyn Laura Kosinsky [3], Yeaeun Han [2,6], Xu Zhang [2,5], Ana Catarina Franco [2,5], Sung Y Kim [6,7], Saranya P Wyles [8], Y S Prakash [5,9], David G Monroe [1,2], Luigi Ferrucci [10], Nathan K LeBrasseur [2,5], Paul D Robbins [11], Laura J Niedernhofer [11], Sundeep Khosla [1,2] ✉ & João F Passos [2,5] ✉

## Abstract

**Senescent cells drive age-related tissue dysfunction via the induction of a chronic senescence-associated secretory phenotype (SASP). The cyclin-dependent kinase inhibitors p21[Cip1] and p16[Ink4a] have long served as markers of cellular senescence. However, their individual roles remain incompletely elucidated, particularly in vivo. Thus, we conducted a comprehensive examination of multiple single-cell RNA sequencing datasets spanning both murine and human tissues during aging. Our analysis revealed that *p21[Cip1]* and *p16[Ink4a]* transcripts demonstrate significant heterogeneity across distinct cell types and tissues, frequently exhibiting a lack of co-expression. Moreover, we identified tissue-specific variations in SASP profiles linked to p21[Cip1] or p16[Ink4a] expression. Using RNA velocity and pseudotime analyses, we discovered that *p21*+ and *p16*+ cells follow independent trajectory dynamics, with no evidence of direct transitions between these two states. Despite this heterogeneity, we identified a limited set of shared "core" SASP factors that may drive common senescence-related functions. Our study underscores the substantial diversity of cellular senescence and the SASP, emphasizing that these phenomena are inherently cell- and tissue-dependent.**

**Keywords** Cellular Senescence; Senescence-Associated Secretory Phenotype (SASP); Heterogeneity; Single-Cell Mapping; Aging
**Subject Categories** Cell Cycle; Methods & Resources; Molecular Biology of Disease

## Introduction

Cellular senescence is characterized by not just an irreversible cell-cycle arrest but also the development of various functional and morphological alterations in distinct cell compartments, such as the nucleus, lysosomes, mitochondria, and others (Gorgoulis et al, 2019; Hayflick and Moorhead, 1961). The senescence-associated cell-cycle arrest is primarily mediated by cyclin-dependent kinase inhibitors (CDKis), notably p21[CIP1] and p16[INK4A]. For simplicity, we refer to the gene transcripts as *p21* and *p16*, and to the proteins as p21 and p16, respectively (Gorgoulis et al, 2019). Although p27 has been reported in skin (Liu et al, 2015) and fibroblasts (Wagner et al, 2001), and p19 in the brain (Dehkordi et al, 2021), these are not universally involved in senescence. Yet, current guidelines by the senescence research community prioritize p21 and p16 as key CDKi for senescence characterization (Ogrodnik et al, 2024; Suryadevara et al, 2024). Senescent cells also exhibit a senescence-associated secretory phenotype (SASP), which consists of a diverse array of secreted factors including immune-modulatory cytokines and chemokines, matrix remodeling enzymes, and growth factors (Coppé et al, 2008). Senescent cells play crucial roles in development, tumor suppression, and tissue repair (Demaria et al, 2014; Muñoz-Espín et al, 2013; Serrano et al, 1997). However, as individuals age, the accumulation of these cells has been linked to the onset of various age-related conditions. Additionally, in mice, the removal of senescent cells either genetically or pharmacologically prevents the development of age-related adverse outcomes, underscoring the therapeutic potential of targeting these cells (Robbins et al, 2021).

Even though there are numerous molecular changes associated with senescent cells, detecting these cells within tissues remains exceedingly challenging. Central to this difficulty is the absence of a singular specific marker for the unequivocal identification of senescent cells, as none of the markers conventionally employed in senescence detection exhibit individual specificity. Adding to this complexity, senescent cells and their SASP exhibit variability

[1]Division of Endocrinology, Mayo Clinic, Rochester, MN 55905, USA. [2]Robert and Arlene Kogod Center on Aging, Mayo Clinic, Rochester, MN 55905, USA. [3]Robert Bosch Center for Tumor Diseases, Stuttgart, Germany. [4]Eberhard Karls University Tuebingen, Tuebingen, Germany. [5]Department of Physiology and Biomedical Engineering, Mayo Clinic, Rochester, MN 55905, USA. [6]Department of Biochemistry, Konkuk University School of Medicine, Seoul, Republic of Korea. [7]Research Institute of Medical Science, Konkuk University, Seoul, Republic of Korea. [8]Department of Dermatology, Mayo Clinic, Rochester, MN 55905, USA. [9]Department of Anesthesiology, Mayo Clinic, Rochester, MN 55905, USA. [10]Intramural Research Program, National Institute on Aging, NIH, Baltimore, MD, USA. [11]Institute on the Biology of Aging and Metabolism, Department of Biochemistry, Molecular Biology and Biophysics, University of Minnesota, Minneapolis, MN, USA. ✉E-mail: dominik.saul@med.uni-goettingen.de; Khosla.Sundeep@mayo.edu; passos.joao@mayo.edu

contingent upon the specific physiological context, stress, cell type, and tissue under investigation (Hernandez-Segura et al, 2017).

The recent advancements in single-cell omics technologies offer a unique opportunity to comprehensively unravel the heterogeneity of the senescent phenotype across various cell types and tissues (Gurkar et al, 2023). One of the key unresolved questions concerns the relative contributions of p16 and p21, which have been identified as critical drivers of cellular senescence, towards age-related senescence across different tissues in vivo. By analyzing multiple in vivo scRNA-seq datasets—including *Tabula Muris Senis* (Almanzar et al, 2020) and the Calico murine aging cell atlas (Kimmel et al, 2019)—across diverse murine tissues (brain, skeletal muscle, bone, and liver) as well as human skin and lung during aging, we found that p16 and p21 expression arises in tissue-specific cell populations. These populations display unique, often non-overlapping secretory profiles, suggesting that they play distinct functional roles.

Furthermore, our findings indicate that while there are commonalities in SASP profiles in *p16* and *p21* expressing cells, these vary considerably according to tissue- and cell-type. In addition, we find that a small number of common SASP markers can be considered as a "core" set associated with cellular senescence. Our comprehensive analysis underscores the intricate nature of cellular senescence and the associated secretory profile, emphasizing the important role of single-cell studies to fully elucidating and characterizing senescence in aging tissues.

# Results

## Unraveling the p16- vs. p21-associated secretome in the murine brain

Previous studies have shown that markers of cellular senescence increase during aging in the murine brain and, importantly, that clearance of *p16+* cells enhances cognitive function in aged mice (Ogrodnik et al, 2021). To further investigate cellular senescence in the brain, we conducted an in-depth analysis of scRNA-sequencing datasets (Ogrodnik et al, 2021) to profile and compare the cellular composition and transcriptomes of young (4 m) and old mouse (24 m) hippocampi; a brain region known for its involvement in memory formation. Our analysis initially identified five primary cell populations within the hippocampus (Fig. 1A). To mitigate potential confounding factors introduced by inflammatory immune cells, we refined our focus by excluding CD45$^{high}$ cells. This criterion still allowed the inclusion of microglia in our analysis, which are characterized by CD45$^{low/intermediate}$ expression (Martin et al, 2017). Further filtering steps involved ensuring the absence of the proliferation marker Ki67 (*mKi67*) and verifying that the selected cells were not in the S phase, since senescent cells are arrested in G1/2 phases of the cell-cycle (Gire and Dulic, 2015). Subsequently, *p16*(*Cdkn2a*)-positive cells and *p21*(*Cdkn1a*)-positive cells were identified. The selection process is depicted with an increasing percentage of G0/G1 cells across each individual step (Fig. 1B). Among the *p16+* cells, microglia and oligodendrocytes appeared as the main subpopulations, whereas in the *p21+* cells, microglia predominated (Fig. 1B). Interestingly, while all *p16+* cells increased with age, the increase of *p21+* cells was restricted to specific clusters (Fig. EV1A,B).

Our further analysis revealed the presence of three subpopulations: (1) cells exclusively expressing *p21*; (2) cells exclusively expressing *p16*; and (3) cells expressing both *p21* and *p16* (Fig. 1C,D). The unbiased trajectory analysis using scVelo and PhyloVelo (describing the rate of gene expression change at a given time point) showed that *p21+* and *p16+* cells do not share a common developmental pathway, indicating that neither cell type arises from the other. This suggests that these populations are distinct and do not transition into one another (Wang et al, 2024b) (Fig. 1E). To gain deeper insights into the composition of the associated secretome within these identified subpopulations, we leveraged our previously established SenMayo gene set (Saul et al, 2022), which has been demonstrated to be commonly regulated across various age-related transcriptome datasets and is primarily composed of SASP-related factors (Brizio et al, 2024; Carapeto et al, 2024; Suryadevara et al, 2024). Interestingly, the variability of expressed SASP factors among *p16+* cells was substantially higher, as demonstrated by their standard error, compared to that of *p21+* cells. This suggests that the secretory phenotype in *p21+* cells may be a more conserved or consistent mechanism—or that *p21+* cells are more homogeneous (Fig. 1F). We observed a distinct secretory profile for *p21+* and *p16+* cells, marked by limited overlap between the two populations. Notably, only two genes, *Cxcl16* and *Plaur*, were expressed in both subpopulations (Fig. 1G). Likewise, an expanded analysis of 1989 secreted protein genes (as listed by the Human Protein Atlas (Uhlén et al, 2015)), which included the secreted proteins plus SenMayo, revealed limited overlap between *p21+* and *p16+* cells. Only four genes, *Col8a2*, *Plaur*, *Cxcl16* and *Fbln5* were shared between the two subpopulations (Fig. EV1C). To eliminate potential cell type-specific biases between *p21+* and *p16+* cells, we focused on microglia and oligodendrocytes, as these represent the cell types with the highest proportion of *p21+* and *p16+* cells (Fig. 1B). We found that *p21+* and *p16+* microglia or oligodendrocytes had distinct secretory profiles, with only partial overlap in microglia, mirroring the differences seen in the overall population. This suggests that these variations are primarily driven by p16 or p21 expression rather than intrinsic cell type differences (Fig. EV1D,E). To further explore the differences between *p21+* and *p16+* cells and refine the RNA velocity analysis, we conducted a biased pseudotime analysis using Monocle3 (Cao et al, 2019), following the initial unbiased PhyloVelo approach. This allowed us to trace the dynamic trajectories of these subpopulations within the brain dataset, providing a more detailed understanding of their developmental pathways. Surprisingly, the results showed that the trajectory dynamics were independent of p16 or p21 expression status, with no evidence of direct lineage progression from *p16+* to *p21+* cells (Fig. EV1F,G).

To validate our initial observations, we analyzed an independent brain dataset from eight young and eight old whole mouse brains (excluding hindbrain (Ximerakis et al, 2019)). As observed in the first dataset, *p21+* and *p16+* cells showed minimal overlap and displayed distinct secretory profiles (Fig. EV2A–E). We also found that the majority of secretory markers were associated with *p21+* cells and closely mirrored the patterns identified in the first dataset (Fig. EV2F). Consistent with our previous results, *p16* expression significantly increased with age in the brain, whereas *p21* upregulation was confined to specific cell clusters (Fig. EV2G,H). However, it is important to highlight that the first dataset was derived from isolated hippocampal tissue, whereas the second

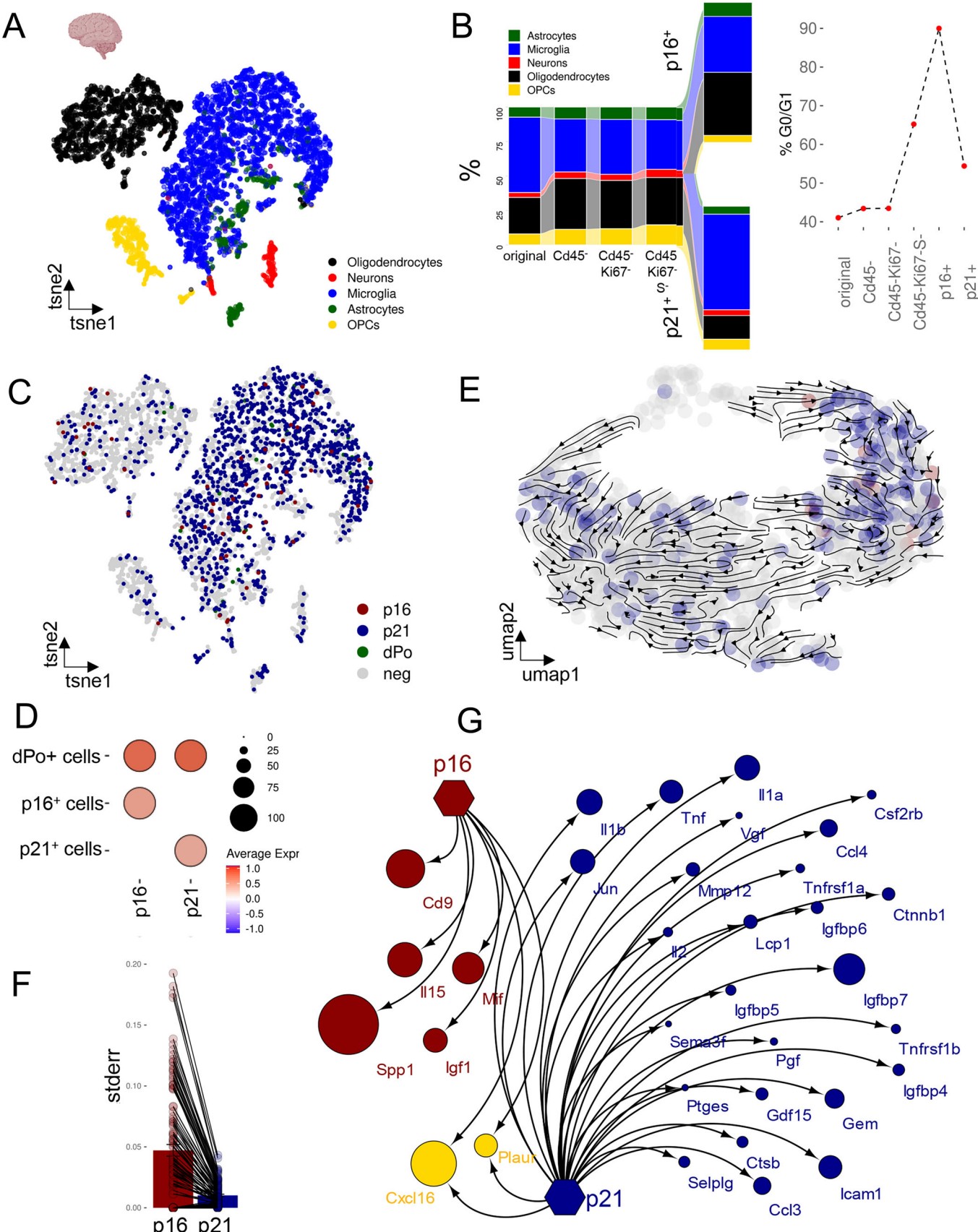

**Figure 1.  Distinct p16+ and p21+ cell populations with differential secretory profiles in the murine hippocampus.**

(A) A t-SNE plot depicting the five main populations of young and old murine hippocampus (Ogrodnik et al, 2021, GSE161340). (B) From the original population, CD45− cells are selected, followed by Ki67-negativity and cells not in the S phase. From these cells, just p16(Cdkn2a)+ cells and p21(Cdkn1a)+ cells are selected. From the p16+ cells, microglia and oligodendrocytes depict the main populations, while in the p21+ population, microglia is predominant. (C) t-SNE visualization of p16 + (red) and p21+ (blue) cells, along with a few double-positive cells (green), shown in the Cd45-Ki67-S- population. (D) The dPo (double positive) cells are high in both Cdkn2a and Cdkn1a, while p16+ cells just express Cdkn2a, but no Cdkn1a and vice versa. (E) Velocity depicts that there is no common ancestor for p16+ cells or p21+ cells, but that these are rather distinct entities. (F) The standard error for SASP factors is substantially higher in the p16 compared to the p21 population. (G) Utilizing the SenMayo gene set, SASP factors exclusively expressed by p16+ cells are fewer than those expressed in p21+ cells, with some (Cxcl16, Plaur) being expressed by both. Very few secretory genes are expressed by p16-negative and p21-negative cells. The size of the dots represents the fold change compared to all other populations shown in (C).

dataset included the entire brain. This distinction may account for some of the observed differences in cell distribution and secretory profiles between the two datasets.

Overall, these data emphasize the diversity as well as variability in $p21+$ and $p16+$ associated secretome profiles among cells with core features of senescence in the brain.

## Comparing p16 and p21-associated secretome across murine tissues

After our initial observations in the brain, we extended our investigation to assess the generality of our findings in diverse murine tissues during aging. We specifically focused on skeletal muscle, bone, and liver, as previous research had indicated age-dependent increases in senescence-associated markers and demonstrated the benefits of eliminating senescent cells for the functional outcomes of these organs (Farr et al, 2017; Ogrodnik et al, 2017; Zhang et al, 2022). We used single-cell RNA-seq datasets comparing young and old mice. In total, we successfully identified 9, 14, and 13 distinct cell populations in skeletal muscle (Fig. 2A), bone (Fig. 2E), and liver (Fig. 2I), respectively. All analyses were repeated in independent datasets from the same tissue, resulting in similar observations across tissues (muscle, bone, and liver were taken from the *Tabula Muris Senis* (Almanzar et al, 2020)) (Figs. EV3–EV5). We then followed a similar methodology as in our brain analysis, excluding cells that were positive for Ki67, in the S-phase of the cell cycle, or expressing CD45high cells.

In our examination of all three tissues (Fig. 2), we observed a consistent pattern of distinct cell subpopulations that expressed either $p21$ or $p16$, with only a rare subset co-expressing both markers (Fig. 2B,F,J). Similar to our findings in the brain, RNA velocity and pseudotime analysis revealed that there was no evidence of a common ancestor between $p21+$ and $p16+$ cells, reflecting distinct developmental trajectories for each subpopulation. Furthermore, $p16+$ cells exhibited greater heterogeneity compared to $p21+$ cells (Fig. 2C,G,K).

In all three tissues, the $p21+$ cell population was considerably more abundant than the $p16+$ population. Using the SenMayo gene set, we found that the pattern observed in the brain was consistent across other tissues: while $p16+$ cells uniquely expressed a small subset of SenMayo factors, $p21+$ cells displayed a broader range of SenMayo genes (Fig. 2D,H,L). Although a small number of genes were co-expressed by cells positive for both $p16$ and $p21$, these were not consistently shared among the three tissues. Indeed, our investigations suggest that there is a distinctiveness in the composition of secretory profiles for each tissue, irrespective of the expression of $p21$ and $p16$. If we extend our analysis to a larger

set of genes, including an extensive list from the Human Protein Atlas (Uhlén et al, 2015) plus SenMayo, we observe a consistent pattern across murine tissues: $p21+$ cells are associated with a larger secretome-associated transcriptional profile, with limited overlap in secreted gene expression between p21* and p16* cells (Fig. EV3A).

We confirmed the generality of our findings by performing the same analysis on separate independent scRNA-seq datasets from muscle (Fig. EV3B–J). Additionally, we assessed whether our analysis of the $p16+$ vs. $p21+$ secretome-associated transcriptional profile remained consistent when focusing on specific cell types. We found that a similar distribution of the $p16+$ vs. $p21+$ secretome was evident in fibro-adipogenic progenitors (FAPs) within skeletal muscle (Fig. EV3), in Lepr+ MSCs in bone (Fig. EV4A–J), and in endothelial cells in the liver (Fig. EV5A–J).

To determine if the two distinct cell populations are also present at the protein level in bone tissue, we analyzed data from mass cytometry by time-of-flight (CyTOF) on bones from young and aged mice (Doolittle et al, 2023). Similar to the scRNA-seq analysis, we excluded cells positive for Ki67 and CD45. Our analysis revealed distinct subsets of cells expressing either p21 or p16 exclusively, along with a rare subgroup co-expressing both markers at the protein level (Fig. EV6A–C). However, it is important to note that CyTOF relies on targeted panels of antibodies, which limited our analysis of secretory profile components. Most importantly, the lack of available SASP components makes it impossible to directly compare these proteomic data to our transcriptomic analyses. Nevertheless, we identified proteins such as Serpine1 exclusively expressed in $p16+$ cells and IL-6 exclusively expressed in $p21+$ cells, suggesting the existence of distinct secretory transcriptional profiles (Fig. EV6A–C). We further analyzed murine spleen (Fig. EV7A–E) and kidney (Fig. EV7F–J) using the Calico dataset (Kimmel et al, 2019). Consistent with our previous observations, $p16+$ and $p21+$ cells remained distinct populations in these tissues, each characterized by a unique secretome profile.

To determine whether the differences observed between $p21+$ and $p16+$ cell populations during aging are specific to age-related senescence or also occur in disease contexts, we compared our results with a mouse model of metabolic dysfunction-associated steatohepatitis (MASH) (Bendixen et al, 2024), a condition characterized by elevated senescence markers (Yashaswini et al, 2024) (Fig. EV8A–C). Similar to an aging liver, MASH showed minimal overlap between $p21+$ and $p16+$ cells (Fig. EV8B). However, the secretome transcriptional profiles associated with $p21$ and $p16$ in MASH were distinct from those observed in the aging liver (Fig. EV8D,E), suggesting that the composition of the

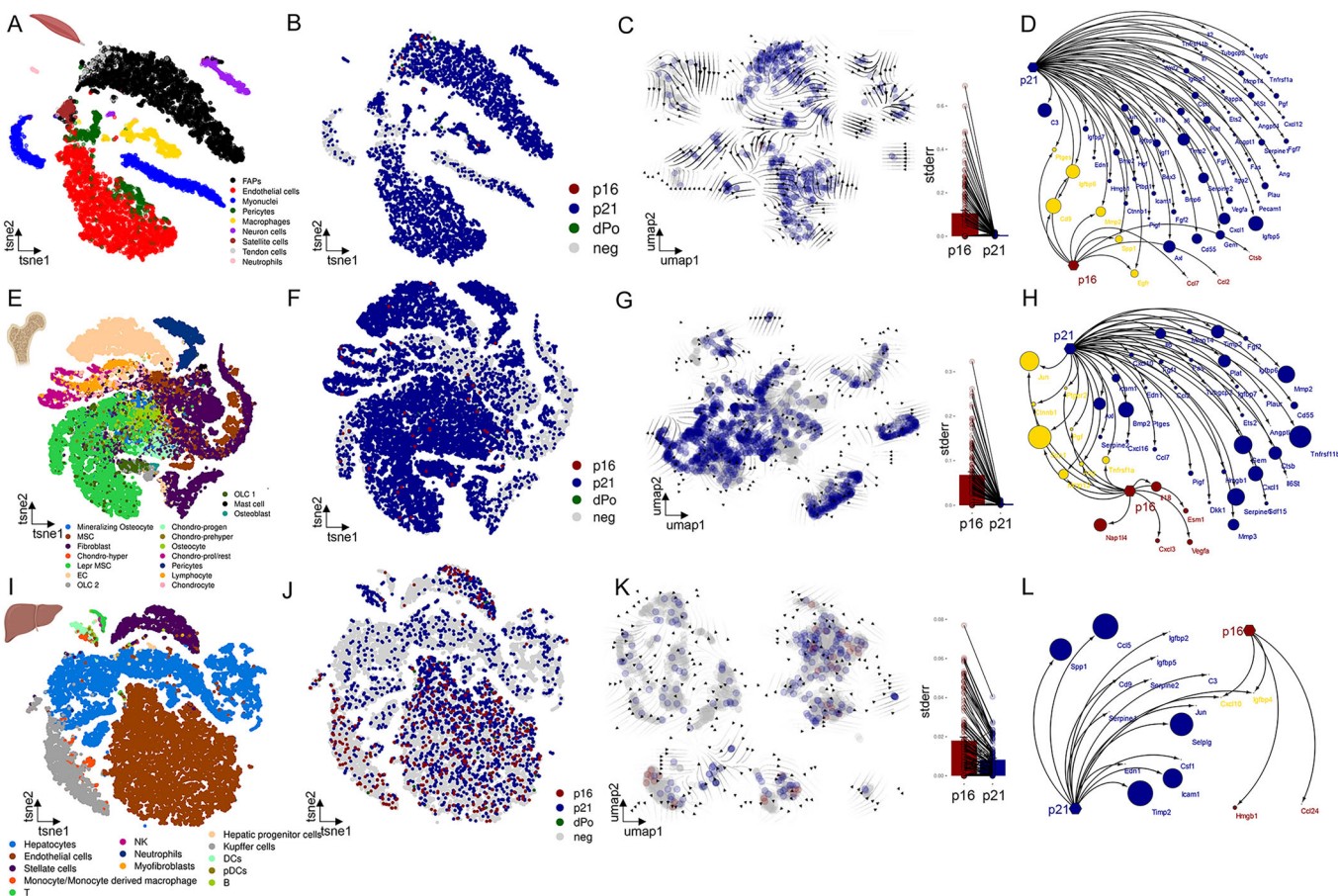

**Figure 2. Comparative analysis of *p16*+ and *p21*+ cells and their secretory phenotype in different murine tissues during aging.**

(A) In murine skeletal muscle (Zhang et al, 2022, GSE172410), nine different cell types can be distinguished. (B) *p21*+ cells constitute the majority, while double positive (dPo) cells are infrequent. (C) The developmental trajectory points to different points of origin for *p16*+- and *p21*+ cells. (D) The secretory phenotype is heterogeneous, with *p21*+ cells expressing a vast array of SASP factors, exhibiting minimal overlap with p16-associated secretory factors. (E) In murine bone (Baryawno et al, 2019, GSE128423), 17 cell types can be distinguished. (F) *p21*+ cells once again dominate the senescent cell population. (G) There is no common ancestor for *p16*+- and *p21*+ cells, and the (H) secretory profile in murine bone remains diverse, with *p21*+ cells expressing a larger number of secretory factors compared to *p16*+ cells. (I) In the murine liver (Su et al, 2021, GSE166504), 13 different cell types are identified. (J) The liver has the lowest proportion of *p21*+ cells from all tissues analyzed, although they still form the majority of senescent cells, with a few dPo cells. (K) There are different origins for *p16*+ and *p21*+ cells in the liver. (L) The secretory phenotype linked to p21 is more extensive than that associated with p16, with minimal overlap between the two.

secretory profile is influenced by the specific stimuli driving senescence in vivo.

Overall, our findings support the hypothesis that *p21*- and *p16*-expressing cells represent distinct senotypes with unique secretory profiles that vary depending on the tissue type and the senescence-inducing stimuli. This is in line with previous studies in which overexpression of p21 or p16 in mouse embryonic fibroblasts led to distinct secretome profiles (Sturmlechner et al, 2021), showing only partial overlap (Fig. EV9A,B).

## Comparing the p16- and p21-associated secretome in human tissues during aging

We next assessed the consistency of our findings across species. To achieve this, we analyzed scRNA-seq datasets from human skin and lung tissues during aging (Jia et al, 2023; Zou et al, 2021). To examine human skin, we utilized a recently published dataset in which scRNA-sequencing was conducted on human eyelid skin

samples from individuals spanning an age range of 18 to 76 years (Zou et al, 2021). Here, we identified 16 distinct cell populations (Fig. 3A). We then proceeded to remove Ki67+, CD45[high] and S-phase cells (Fig. EV10A,B) and observed three subpopulations: *p21*-exclusive, *p16*-exclusive, and *p21*-*p16* co-expressing cells, with *p21*-exclusive being the predominant group (Fig. 3B). Similar to our previous findings, there was no common origin of *p16*+ and *p21*+ cells, and the secretome composition of *p16*+ cells was more heterogeneous than the composition of *p21*+ cells (Fig. 3C). Interestingly, by utilizing the SenMayo dataset, we observed no overlap between *p21*+ and *p16*+ cells, underscoring their distinct identities (Fig. 3D), which was also found in a broader secretory panel (Fig. EV10C). The analysis of keratinocytes demonstrated that the distinct secretome compositions in *p21*+ and *p16*+ cells are not determined by the cell type itself (Fig. EV10D).

To further substantiate our mRNA-based findings, we performed immunofluorescence imaging of sun-protected aged human

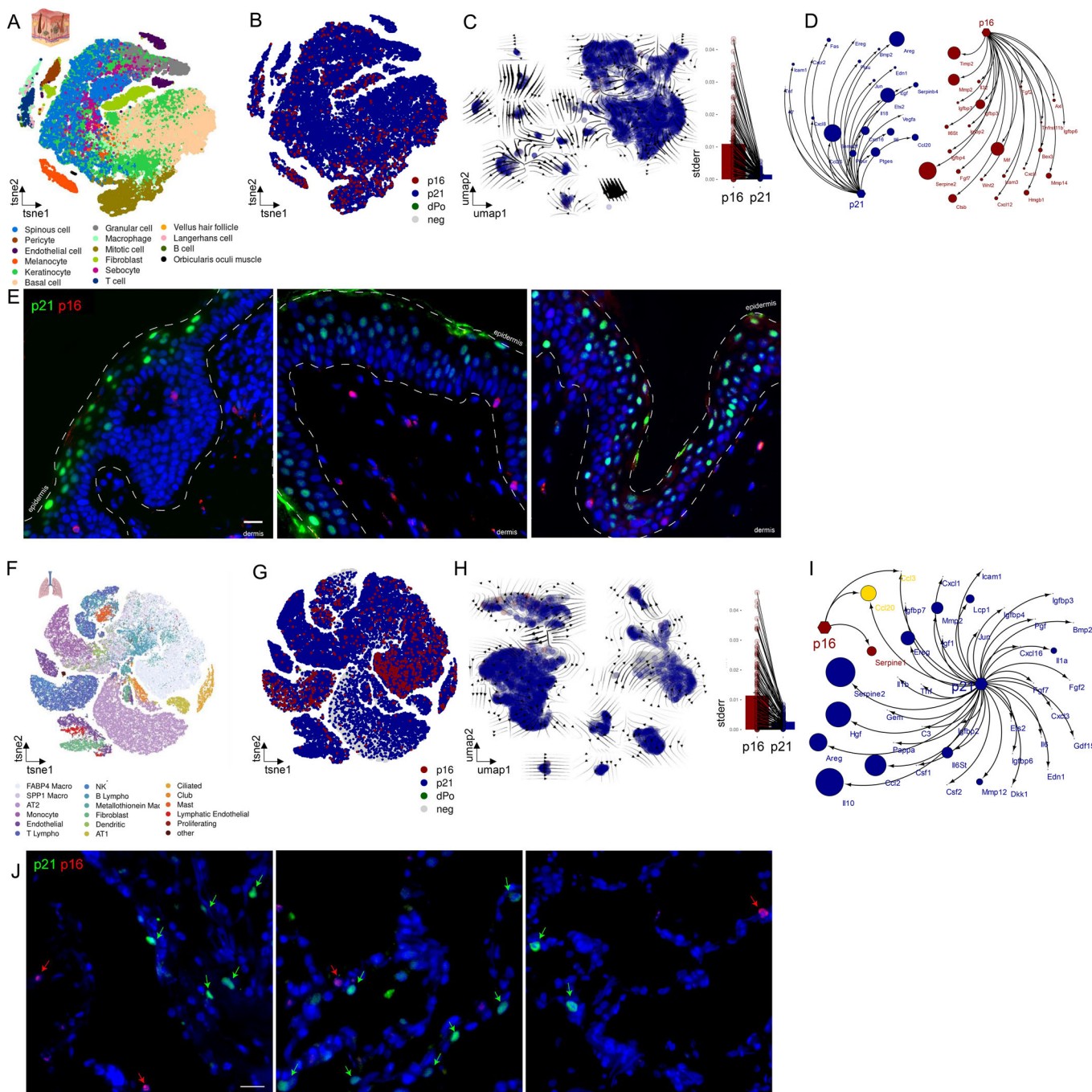

**Figure 3. Comparative analysis of *p16+* and *p21+* cells and their secretory phenotype in human skin and lung during aging.**

(A) In human skin samples (Zou et al, 2021, HRA000395), 16 distinguishable cell populations are identified. (B) Among these populations, *p21+* cells are notably abundant, while double-positive (dPo) cells are rare. (C) RNA-Velocity reveals a distinct pattern for *p16+* and *p21+* cells, and a higher heterogeneity regarding the secretory factor expression in *p16+* cells. (D) The secretory profile expressed by *p21+* cells is significantly larger compared to the secreted factors associated with *p16+* cells. (E) Immunofluorescence analysis on aged human skin depicting *p16+-* (red) and *p21+* (green) cells display distinct entities. Scale bar: 20 μm. (F) In the human lung (Jia et al, 2023), GSE122960, GSE128033, GSE130148, and GSE212109, 18 discernible cell types are identified, and (G) within these cell types, *p21+* cells constitute the majority of senescent cells. (H) RNA-Velocity reveals no common ancestor of *p16+* and *p21+* cells, while *p16+* cells exhibit a more diverse secretory expression profile. (I) The secretory profile of *p21+* cells is extensive compared to that of *p16+* cells, with minimal overlap. (J) In human lung tissue (COPD and pulmonary fibrosis/IPF), similarly to aged skin tissue, *p21+* (green) and *p16+* (red) cells display distinct entities. Scale bar: 20 μm. Source data are available online for this figure.

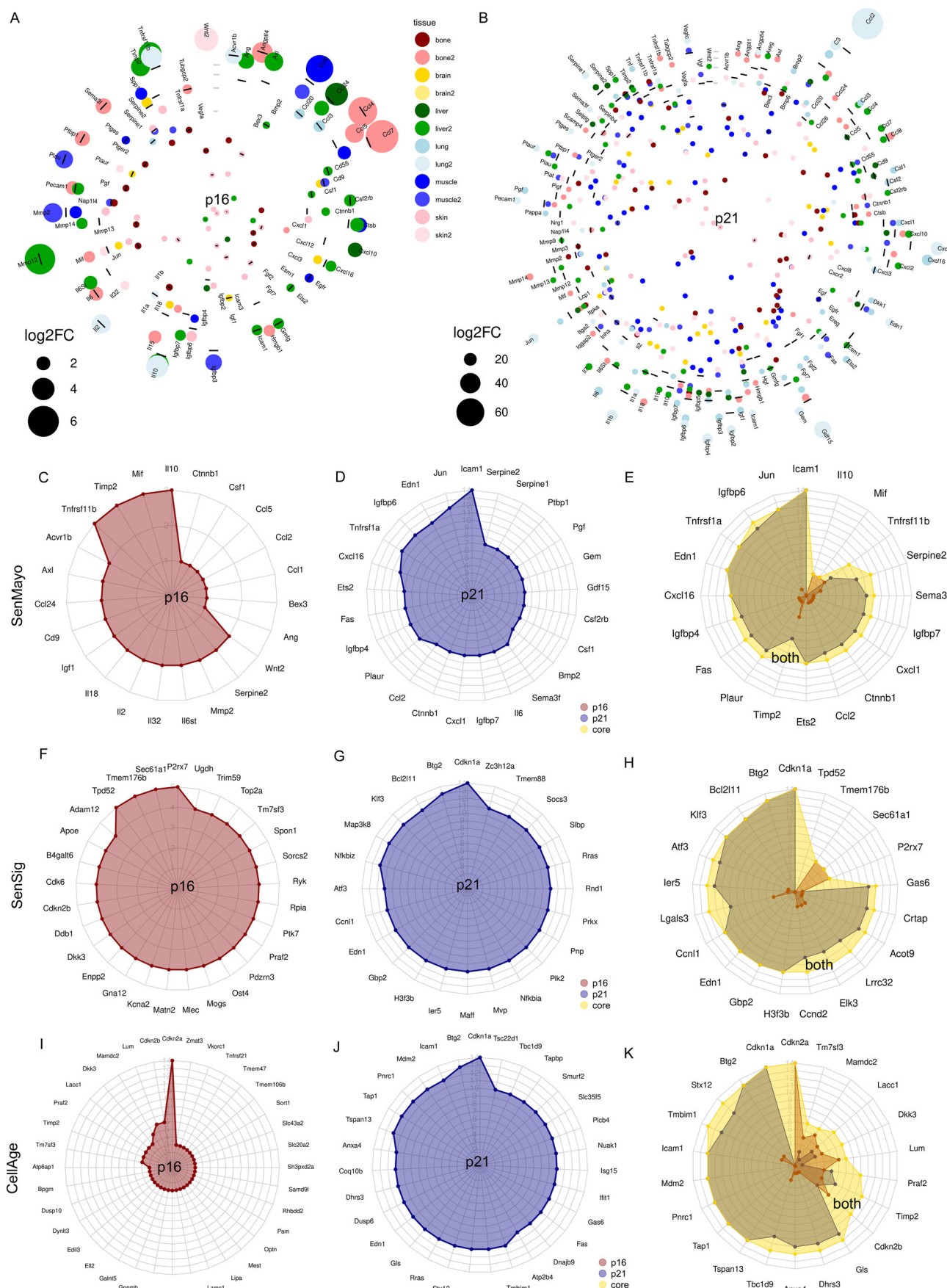

**Figure 4. Common p16- and p21-associated secretory phenotype, SenSig, and CellAge across murine and human tissues.**

(A) The p16-associated secretory phenotype is characterized by the predominant presence of *Ccl2, Ccl4*, Ccl7, Mmp12, and *Ccl24*, with some markers, such as *Tnfrsf11b*, expressed in multiple tissues. (B) The p21-associated secretory phenotype displays greater diversity, with genes like *Ccl2, Cxcl12, Cxcl16*, and *Jun* commonly expressed across tissues. (C) From the SenMayo panel, the p16-core-secreted factors include *Il10, Mif, Timp2*, and *Tnfrsf11b*. (D) The p21-associated secreted factors are dominated by *Icam1, Jun*, and *Edn1* along with *Igfbp6* and *Tnfrsf1a* that are expressed consistently by *p21*+ cells in at least 11 of the 14 analyzed tissues. (E) A "core"-secreted phenotype for senescent cells comprises *Icam1, Jun, Igfbp6, Tnfrsf1a*, and *Edn1*. (F–H) The significantly associated genes with p16 and p21, respectively, from SenSig (Cherry et al, 2023) and (I–K) CellAge (Avelar et al, 2020) gene sets. The depicted genes are those significantly overexpressed (*p* < 0.05, FC >0), with cell type colors corresponding to those in Figs. 1–3. In (A, B), the dot size reflects the log2FC, the circular y-axis is in log$_{10}$- scale. For (C) only markers expressed in one or more tissues are shown; in (D) those expressed in eight or more tissues; in (F) those expressed in four tissues; and in (G), those expressed in more than 11 tissues are displayed.

skin (Fig. 3E), which confirmed at the protein level, distinct populations of p16⁺ and p21⁺ cells (Fig. 3F).

We independently confirmed these findings using another human skin dataset (Solé-Boldo et al, 2020) (Fig. EV10E–J). Because all our previous observations were derived from scRNA-seq, we sought to validate these findings using an independent transcriptomic approach that also preserves spatial information. This was particularly important, as cell dissociation required for scRNA-seq may result in the loss of fragile senescent cells. Applying MERFISH to human skin confirmed that p16⁺ and p21⁺ cells represent distinct populations across different skin cell types, each with a divergent SASP profile and only minimal overlap (Fig. EV10K–M).

A similar pattern was observed in the human lung. Here we utilized a recently published scRNA-seq dataset from healthy human lungs ranging from 21 to 78 years of age (Jia et al, 2023). Within our analysis, we identified 18 distinct cell populations (Fig. 3F). After excluding Ki67+, CD45$^{high}$ and S-phase cells (Fig. EV11A–C), we observed that a significant majority of cells displayed *p21* expression, while only a small number of cells exhibited *p16* or the combination of *p16* and *p21* (Fig. 3G). There was no common ancestor, as demonstrated by RNA-velocity, and p16+ cells exhibited a heterogeneous secretome expression (Fig. 3H). Notably, cells that were *p16*+ demonstrated limited expression of SenMayo components, with *Serpine1* being the exclusive gene expressed by this subgroup. In contrast, *p21*+ cells exhibited the expression of multiple SenMayo components (Fig. 3I). In addition, we analyzed alveolar type II (AT2) cells and again found distinct secretome transcriptional profiles associated with *p21* and *p16* (Fig. EV11D).

In an independent experiment, we analyzed human diseased lung tissue (COPD and pulmonary fibrosis/IPF) by immunofluorescence, which confirmed at the protein level the presence of distinct p16⁺ and p21⁺ cell populations (Fig. 3J).

We independently validated our findings in a murine lung dataset from the Calico dataset (Kimmel et al, 2019) (Fig. EV11E–J). When we broaden our examination to include a wider range of genes which involves a comprehensive list of secreted proteins (Uhlén et al, 2019) alongside SenMayo (Saul et al, 2022), we observed that there are relatively few secreted protein genes that are common between *p21* and *p16* expressing cells in both human and murine lung (Fig. EV11A,I,J). This further underscores the functional specificity inherent in these subpopulations.

## Common p16 and p21 -associated secretomes across murine and human tissues

Following our identification of significant heterogeneity in the secretome profiles expressed between *p16*+ and *p21*+ cells among

tissues in both murine and human samples, we sought to investigate which elements exhibited commonality across diverse tissues and species (Fig. 4A,B). When investigating the *p16*+ SenMayo gene set, we noticed that only a few SenMayo genes were expressed in two or more of the analyzed tissues. This subset included *IL-10, Mif, Timp2, Tnfrsf11b, Acvr1b, Axl, Ccl24, Cd9, Igf1, IL-18, IL-2, IL-32, IL-6st, Mmp2, Serpine2*, and *Wnt2* that were expressed in at least two datasets. (Fig. 4C).

In the case of *p21*+ cells, we observed that *Icam1* was consistently expressed across all six analyzed tissues, while several other factors, including *Jun, Edn1, Igfbp6, Tnfrsf1a, Cxcl16, Ets2, Fas, Igfbp4, Plaur*, and *Ccl2* were expressed in eight or more of the 14 analyzed tissues (Fig. 4D). Combining *p16* (red) and *p21* (blue) markers that are expressed in more than one (*p16*) and five (*p21*) tissues, a "core" secretome profile comprised of *Icam1, Jun, Igfbp6, Tnfrsf1a, Edn1, Cxcl16, Igfbp4, Fas*, and *Plaur* is established. While *Icam1* may exhibit a stronger association with p21, *Igfbp4* and *Igfbp6* are prominently expressed by both cell types, suggesting these as "common" *p16/p21*-associated secretory factors (Fig. 4E).

It should be noted that SenMayo mainly consists of secreted genes (c.a. 73%), while SenSig just has 265 secreted genes (c.a. 15%, based on the Human Protein Atlas), and CellAge has 78 secreted genes (c.a. 15%, based on the Human Protein Atlas). In our search for a common *p21*+ and *p16*+ secretome signature, we broadened our analysis to encompass a greater number of secreted factors by using SenSig (Cherry et al, 2023). This approach enabled us to identify 29 genes expressed in *p16*+ cells that were consistently present in (at least) four of the analyzed tissues (Fig. 4F). In *p21*+ cells, we identified 24 distinct genes expressed in a minimum of 11 tissues, and a single gene, *Cdkn1a*, was expressed in all analyzed tissues (Fig. 4G). Upon combining all factors to identify a "common" senescence-associated phenotype, we observed that *Cdkn1a, Btg2, Bcl2l11, Klf3, Atf3*, and *Ier5* remain abundant primarily in *p21*+ cells. Additionally, *Cdkn2b* is expressed in four out of five tissues by *p16*+ cells (Fig. 4H). Using CellAge (Avelar et al, 2020) as reference gene set, the p16-associated genes were (next to Cdkn2a) Cdkn2b and Lum in 6 of the 14 datasets (Fig. 4I). The *p21*-associated genes were (next to Cdkn1a) Btg2, Icam1, Mdm2, Pnrc1, Tap1 and Tspan13, as expressed in 12 of the 14 datasets (Fig. 4J). The "core" secretory phenotype was (after *Cdkn2a* and *Cdkn1a*) mostly consisting of Btg2, Stx12 and Tmbim1 in 13 of the 14 datasets (Fig. 4K).

To further validate our findings, we leveraged the p21-Cre mouse model (Wang et al, 2024a) in which p21-positive cells were GFP-tagged in murine visceral fat tissue (Fig. EV12A,B). Using this model, we confirmed that GFP/p21-positive cells exhibited a significant enrichment of the previously identified *p21*-associated

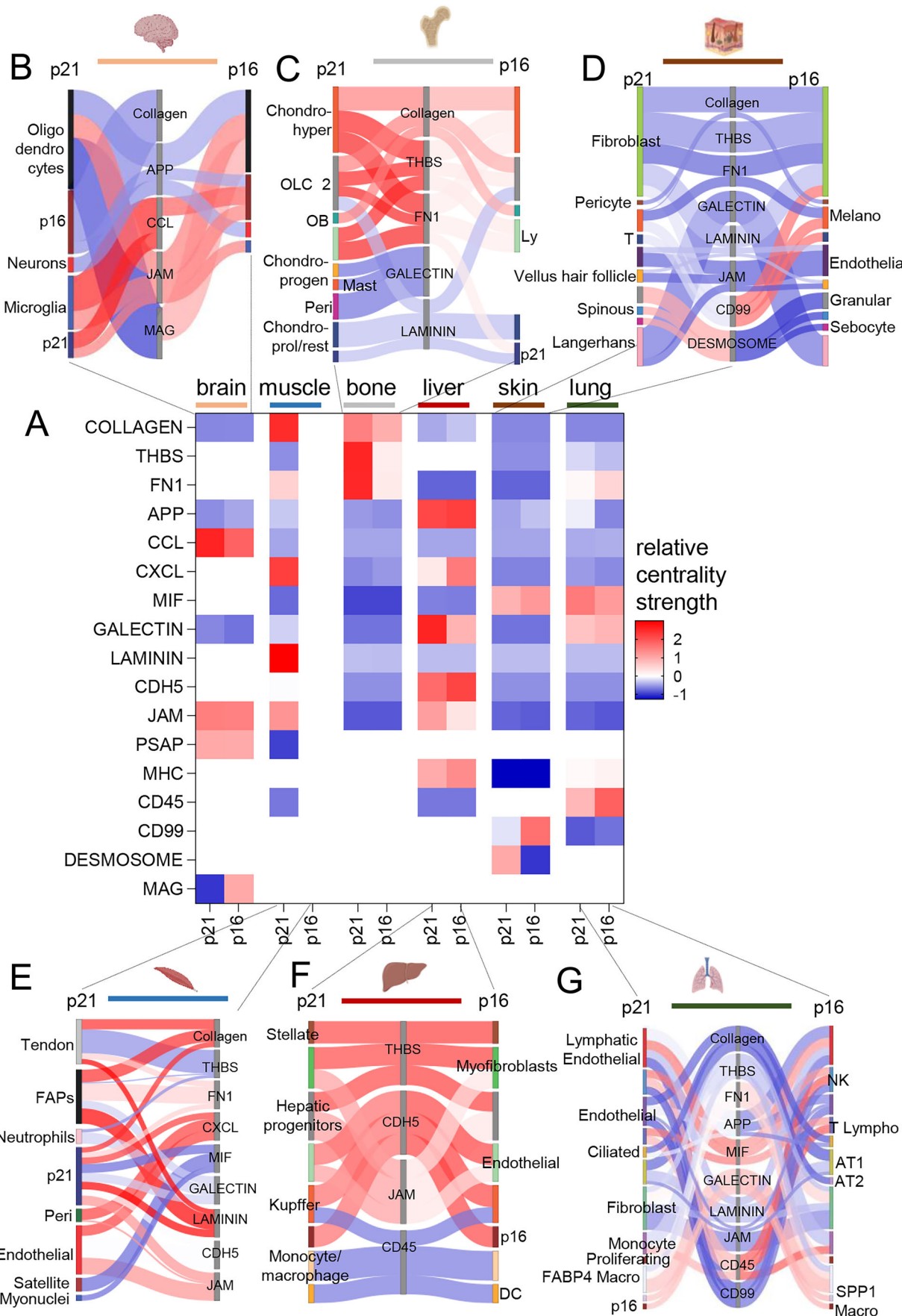

◀

**Figure 5. Unequal communicational patterns between *p21*+ and *p16*+ cells across tissues.**

(A) The network centrality scores depict low (blue) and high (red) communication networks for seventeen significant interactions, with *p21*+ and *p16*+ cells shown pairwise in each tissue. (B) In the brain, *p21*+ cells exhibit higher interaction strength in CCL communication, primarily to microglia cells, while *p16*+ cells demonstrate increased secretory activity via MAG to oligodendrocytes. (C) In bone, *p21*+ cells exclusively utilize the THBS and FN1 pathways to communicate with hypertrophic chondrocytes, osteoblasts, and lymphocytes. (D) In the skin, *p16*+ cells show higher interaction strength in the CD99-pathway to fibroblasts and melanocytes, while *p21*+ cells predominantly signal through the DESMOSOME-pathway to granular and spinous cells, including sebocytes. (E) In muscle, *p21*+ cells mainly signal through the COLLAGEN and LAMININ pathways to fibro-adipogenic progenitors (FAPs) and tendon cells, as *p16*+ cells are too few for calculating communicational patterns. (F) In the liver, the JAM pathway is predominantly used by *p21*+ cells to communicate with myofibroblasts and hepatic progenitors, whereas *p16*+ cells primarily employ the CDH5-pathway to interact with endothelial and Kupffer cells. (G) In the lung, *p16*+ cells exhibit the highest signaling strength in the CD45 pathway, mainly used to communicate with NK cells and T lymphocytes, while *p21*+ cells primarily use the MIF pathway for contact with lymphatic endothelial and NK cells. Displayed are only significant interactions (padj < 0.05), with cell type colors corresponding to those in Figs. 1–3. The color scale (blue-white-red) of the interactions in (B–G) corresponds to the centrality score in (A).

SenMayo, SenSig, and CellAge scores in GFP/p21-positive cells (Fig. EV12C–G).

## Heterogeneous intercellular communication highlights the functional diversity of *p21*+ and *p16*+ cells

The *p21*+ and *p16*+ associated secretome plays a crucial role in intricate intercellular communication, exerting complex effects on neighboring cells. These effects include the propagation of senescence (Acosta et al, 2013), modulation of tissue repair processes (Demaria et al, 2014; Moiseeva et al, 2023; Saul et al, 2021), and recruitment of immune cells (Lagnado et al, 2021). We sought to determine if senescence subtypes (*p21*+ and *p16*+ cells) not only differed in their transcriptomes, but also in how they communicated with other cells. Thus, we utilized CellChat, a commonly used tool for inference of cell-cell communication (Jin et al, 2021). Our analysis focused on both *p21*+ and *p16*+ cells and their interactions with neighboring cell types across the mentioned tissues. An initial pairwise examination of the communication patterns of these specific *p21*+ or *p16*+ cells within each tissue showed a heterogeneity for the most important 17 signaling pathways (Fig. 5A). A more detailed examination in each tissue showed that in brain (Fig. 5B), *p21*+ cells mostly used the CCL and JAM pathway for communication with microglia, while *p16*+ cells favored the MAG-pathway to communicate with oligodendrocytes, consistent with MAG's role in neuron-glial communication (McGonigal et al, 2019). In bone (Fig. 5C), *p21*+ cells used the THBS- and FN1-pathway exclusively to communicate with hypertrophic chondrocytes and osteolineage cells, while *p16*+ cells utilized other mechanisms. In skin (Fig. 5D), the CD99-pathway was favored by *p16*+ cells and the DESMOSOME-pathway by *p21*+ cells to communicate with granular and spinous cells as sebocytes. Muscle (Fig. 5E) was characterized by a low number of *p16*+ cells and a high secretory pattern of the *p21*+ cells, which mostly used the LAMININ pathway to communicate with fibro-adipogenic progenitors (FAPs) and tendon cells. In liver (Fig. 5F), *p21*+ cells favored the JAM pathway to communicate with myofibroblasts and hepatic progenitors, while *p16*+ cells used the CDH5-pathway for communicate with endothelial and Kupffer cells. In the lung (Fig. 5G), the FN1-pathway was more used by the *p16*+ cells, while *p21*+ cells favored the MIF pathway. These findings emphasize that the diversity of the p21 and p16-dependent associated secretome across different tissues reflects the intricate nature of intercellular communication.

This heterogeneity is further reinforced when exploring the transcription factors that control gene expression in *p16*+ and *p21*+ cells. To achieve this, we employed SCENIC, a computational method that allows the prediction of interactions between transcription factors and target genes based on single-cell RNA-seq data (Aibar et al, 2017). SCENIC analyses reveal that transcription factors regulating gene expression in *p16* (Fig. EV13A,B) and *p21* (Fig. EV13C,D) positive cells were mostly tissue-specific, with minimal overlap between tissues. This implies that the secretory phenotype is not exclusively specific to *p16*+ or *p21*+ cells; rather, these cell subtypes are further transcriptionally regulated by factors specific to the tissue in which they are situated.

In summary, the diversity in cellular communication and transcriptional regulation between *p16*+ and *p21*+ cells within the unique environments of the six analyzed tissues underscores their heterogeneity and suggests potentially distinct functions.

## Discussion

p21 and p16 are both associated with the induction of senescence, but they do not always occur together in senescence. Multiple in vitro studies have shown that their presence and expression can vary depending on the senescence-inducing stimuli, the cell type, and the specific context (Maciel-Barón et al, 2016).

p21 is a direct target of the tumor suppressor protein p53. When DNA damage or other stressors occur, p53 becomes activated and binds to the p21 promoter, leading to increased p21 expression. p21, in turn, inhibits the activity of cyclin-dependent kinases (CDKs), halting the cell cycle and promoting senescence. p16, on the other hand, acts through the Retinoblastoma (Rb) pathway (Serrano et al, 1993). p16 inhibits CDK4 and CDK6, which are responsible for phosphorylating Rb. When Rb is not phosphory-lated, it remains active and prevents the cell from progressing through the cell cycle. Both pathways are intricate as they involve numerous upstream regulators and downstream effectors, as well as the presence of diverse side branches. Furthermore, these pathways exhibit substantial interconnections and crosstalk (Rayess et al, 2012).

p16 and p21 are the most commonly used markers for identifying senescent cells and have been highlighted as key indicators in recent consensus guidelines for detecting cellular senescence (Ogrodnik et al, 2024; Suryadevara et al, 2024). Their bulk expression has been extensively utilized to identify senescent cells in different tissues affected by aging or other pathological

conditions (Gorgoulis et al, 2019). However, only recently, with the advancement of single-cell omics technologies, can we truly examine the full extent of their heterogeneity in vivo (Cohn et al, 2023).

Our study clearly demonstrates that, in various aging tissues, cells expressing p21 and p16 at both the transcript and protein levels predominantly constitute separate subpopulations characterized by mRNA expression of distinct secretomes. This suggests the activation of these two pathways may result in functionally diverse consequences and is reflected in the engagement of different intercellular communication pathways.

Some cells may be at different stages of senescence, with p21 being activated earlier in response to stress and p16 appearing at later stages, as observed in vitro (Stein et al, 1999). However, RNA velocity and pseudotime analyses across different tissues reveal that *p21+* and *p16+* cells follow independent trajectories without direct transitions between these states. Developing animal models that allow for the continuous tracking of p21 and p16 expression over time would provide crucial experimental validation for our findings regarding their distinct roles and origins.

Consistent with the functional diversity of these subpopulations, a recent study has demonstrated that p21 and p16 overexpression elicit distinct secretory phenotypes, with the p21-driven phenotype promoting immunosurveillance (Sturmlechner et al, 2021). p21 upregulation has also been suggested as a mechanism that enables senescent cells to resist apoptosis, thereby facilitating their retention in tissues (Yosef et al, 2017). Moreover, recently developed transgenic models where either *p21+* or *p16+* cells can be cleared have been shown to have different functional outcomes. For example, selective elimination of *p21+* senescent cells, as opposed to p*P16+* cells, effectively prevents radiation-induced bone loss (Chandra et al, 2022). Clearance of *p16+* cells has been shown to extend median lifespan (Baker et al, 2016) while p21+ cell clearance has been recently shown to extend both median and maximum lifespan, further indicating distinct biological roles during aging (Wang et al, 2024a).

Interestingly, the composition of the secretome in cells expressing p21 and p16 exhibited significant variation across different tissues, with only a limited number of common factors. This observation holds significant conceptual implications. It indicates that the phenotypes resulting from the activation of senescence-associated pathways during aging are strongly influenced by the specific cell type involved. Moreover, it raises the intriguing possibility that distinct intrinsic mechanisms may contribute to senescence in various tissue types during aging. Finally, it suggests that analysis of SASP components in tissues should be comprehensive in nature and conducted at single-cell resolution. Considering the factors identified from the SenMayo dataset, which are most commonly associated with both p21 and p16-expressing cells, we found ample evidence in the literature supporting their increased expression or secretion by senescent cells and during aging. For instance, *ICAM1* is upregulated in senescent cells (Gorgoulis et al, 2005). Similarly, *IGFBP4* and *IGFBP6* show increased levels in aging tissues and senescent cells (Coppé et al, 2008; Moiseeva et al, 2023), while *EDN1* (Endothelin-1) expression increases in senescent vascular cells (Alcalde-Estévez et al, 2020). Additionally, *CXCL16* is upregulated in aging and inflammatory conditions linked to senescence, and *PLAUR* (uPAR)

is associated with extracellular remodeling and increased secretion by senescent cells (Amor et al, 2020).

Our findings on cell-cell communication patterns align with those reported by Lagger and colleagues, who combined the Tabula Muris Senis with the Calico murine aging cell atlas (Lagger et al, 2023). In our CellChat analyses, we observed a reduction in collagen signaling across four tissues, which was similarly identified in their study. Likewise, the consistent reduction of App in our analyses was highlighted in male tissues and connected to a range of tissue-specific diseases in their work.

Our SCENIC analyses identified several key regulators for p16- or p21-positive senescent cells. Notably, the regulation of p16-associated senescence by SOX5 aligns with findings from Tchougounova and colleagues, who demonstrated that SOX5 induces acute cellular senescence in the brain (Tchougounova et al, 2009). Similarly, we identified JunD and KLF4 as regulators of p21-associated cellular secretion, consistent with previous findings (Li et al, 2002; Chew et al, 2011).

While our study provides valuable insights into the distinct roles of p21 and p16 in different tissues, it is important to acknowledge the limitations of relying on scRNA-seq datasets. Single-cell RNA sequencing is powerful for capturing transcriptomic heterogeneity at a single-cell level, but it is constrained by technical biases such as drop-out effects and limited detection of low-abundance transcripts, which may overlook key components. These limitations suggest that integrating additional modalities, such as single-cell proteomics or spatial transcriptomics, could provide a more complete picture of the functional specialization of senescent cells. Taking this into consideration, we confirmed the lack of co-localization between p16 and p21 at the protein level using immunofluorescence and validated these findings with MERFISH, a spatial transcriptomics technique that preserves tissue architecture and avoids artifacts from cell dissociation. These complementary approaches support the conclusion that p16+ and p21+ cells represent distinct populations. Nevertheless, our study has not yet addressed SASP heterogeneity using spatial proteomics, emphasizing the need for broader analyses in future work.

These limitations notwithstanding, our findings do hold significant implications for the implementation of senolytic therapies in clinical contexts. They underscore the importance of adopting a context-specific approach, not just taking into consideration the subtype of senescent cells, but also the tissue which is being targeted.

## Methods

**Reagents and tools table**

| Reagent/resource | Reference or source | Identifier or catalog number |
|---|---|---|
| **Experimental models** | | |
| **Recombinant DNA** | | |
| **Antibodies** | | |
| P16 | Ventana | 725-4793 |
| P21 | Cell Signaling | 2947S |

| Reagent/resource | Reference or source | Identifier or catalog number |
|---|---|---|
| **Oligonucleotides and other sequence-based reagents** | | |
| CDKN2A probe | Vizgen | ENST00000579755 |
| **Chemicals, enzymes and other reagents** | | |
| NGS | Agilent-Dako | X090710-8 |
| Citrate buffer | Agilent-Dako | S236984 |
| ProLong Gold Antifade Mountant | Invitrogen | P10144 |
| **Software** | | |
| RStudio | Posit PBC | |
| Cytoscape v3.1.0 | http://www.cytoscape.org | |
| MERSCOPE visualizer (2.3) | Vizgen, Inc. | |
| **Other** | | |

Additional details of reagents and software used are provided in the Reagents and tools table.

## Single-cell analysis

The scRNA-seq data were aligned and quantified using the 10X Genomics Cell Ranger Software Suite (v6.1.1) against the murine reference genome (mm10) and human reference genome (hg19). The Seurat package (v4.3.0.1 and 5.0.0) (Butler et al, 2018; Stuart et al, 2019) was used to perform integrated analyses of single cells. Genes expressed in <3 cells and cells that expressed <200 genes and >20% mitochondria genes were excluded from downstream analysis in each sample. The datasets were SCTransform-normalized, and the top 3000 highly variable genes across cells were selected. The datasets were integrated based on anchors identified between datasets before principal component analysis (PCA) was performed for linear dimensional reduction. After normalization and scaling, a shared nearest neighbor (SNN) graph was constructed to identify clusters in the low-dimensional space (top 30 statistically significant principal components, PCs). An unbiased clustering according to the recommendations of the Seurat package was used, if not provided by the authors. The cell types were assigned according to the authors' recommendations or provided metadata. The alluvial plots were designed with the ggalluvial package (v0.12.5). For the DimPlots, the RNA slot was used, and every value above 0 was counted as "positive". After normalization and scaling, we proceeded with the differentially expressed genes analyses.

The differentially expressed markers were identified by the FindMarkers function (ident.1 was specified, and differences calculated to all other clusters) and the Wilcoxon signed-rank test. We used the SenMayo gene set ($n = 125$) or secreted proteins, obtained by the human protein atlas, augmented by the SenMayo secreted SASP factors ($n = 1989$ (Uhlén et al, 2019)) to select the SASP factors or secreted proteins. The cytoscape bubble plots were designed with cytoscape (v3.1.0), and the size of each bubble is proportional to the avg_log2FC. The circular plots were designed with ggplot2 (v3.4.4). The spider plots were designed with the package fmsb (v0.7.5). The circle size is proportional to the log2FC compared to all other clusters, while the color codes the respective

tissue. The bars show the median of all tissues in which the respective gene is upregulated.

RNA Velocity analyses were performed using scVelo as an unbiased measurement of velocity. Here, unsliced and spliced mRNAs are used to predict cellular states, given that the unspliced counts precede the spliced counts (La Manno et al, 2018). After the direction of the velocity had been obtained, a depiction of the trajectory has been done via PhyloVelo, more closely translating the transcriptional dynamics of splicing kinetics with a likelihood-based, more dynamical model (Wang et al, 2024b). In some tissues, these analyses are further amended by a graphical representation with monocle3 (Cao et al, 2019).

Regarding the heterogeneity analyses, the standard error was calculated as standard deviation, divided by the square root of the sample size for each of the SenMayo SASP factors, separated by p16- or p21-positivity.

For the intercellular communication heatmap, we first calculated the respective intercellular communication via CellChat (v1.6.1). The centrality values were extracted and used for the central heatmap, for which we used GraphPad Prism (Version 9.0). The probability of the inferred communication was used for the sankey-network. The depiction of the sankey-network was done via the networkD3 package (v0.4) and exported via jsonlite (v1.8.7). Since muscle had too little *p16+* cells for the CellChat analyses, these were excluded.

For the SCENIC analyses, we used the standard settings (v.1.3.1). For plotting the most important factors, the relative activity for each transcription factor above 1 was chosen for *p21+* and *P16+* cells, respectively, *per* tissue. Since the liver has very few *p16+* cells for proper calculation with SCENIC, these were excluded. For the pie charts, the sum of transcription factors sharing a tissue was calculated, and the respective percentage is demonstrated within the plot.

Table 1 provides an overview of all datasets analyzed in this study, detailing the organ of origin, species, GEO accession number (GSE), and the methodology used for data generation.

## Cytometry by time-of-flight (CyTOF) analysis

The provided fcs-files were read into R by the flowCore package (v2.8.0) and transformed into a Seurat object. All subsequent analyses followed the standard Seurat procedure as described above.

## RNA-sequencing analysis

The fastq files were mapped to the murine genome (mm10), and analysis was performed using the DESeq2 package (v1.38.3) as previously described (Saul et al, 2022). Significantly differentially regulated genes were selected by a Benjamini–Hochberg adjusted $p$ value <0.05 and log2-fold changes above 0.5 or below −0.5.

## Immunofluorescence staining of FFPE tissue sections

Formalin-fixed, paraffin-embedded (FFPE) tissue sections (5 μm) were prepared from three sun-protected aged skin biopsies from individuals aged 73, 79, and 83 years, and two lung tissue sections from 75-year-old individuals diagnosed with IPF and with COPD, respectively.

**Table 1.** **Summary of datasets used in this study.**

| Figure | Organ | Species | GSE | PMID | Methodology |
|---|---|---|---|---|---|
| Fig. 1 | Hippocampus | *Mus musculus* | GSE161340 | 33470505 | scRNA-Seq |
| Fig. 2a–d | Skeletal muscle | *Mus musculus* | GSE172410 | 36147777 | scRNA-Seq |
| Fig. 2e–h | Bone | *Mus musculus* | GSE128423 | 32103177 | scRNA-Seq |
| Fig. 2i–l | Liver | *Mus musculus* | GSE166504 | 34755088 | scRNA-Seq |
| Fig. 3a–d | Skin | *Homo sapiens* | HRA000395 | 36049540 | scRNA-Seq |
| Fig. 3f–i | Lung | *Homo sapiens* | GSE122960 | 30554520 | scRNA-Seq |
| | Lung | *Homo sapiens* | GSE128033 | 31221805 | scRNA-Seq |
| | Lung | *Homo sapiens* | GSE130148 | 31209336 | scRNA-Seq |
| | Lung | *Homo sapiens* | GSE212109 | 37706427 | scRNA-Seq |
| Ext. Fig. 2 | Brain | *Mus musculus* | GSE129788 | 31551601 | scRNA-Seq |
| Ext. Fig. 3e–i | Skeletal muscle | *Mus musculus* | GSE132042 | 32669714 | scRNA-Seq |
| Ext. Fig. e–i | Bone | *Mus musculus* | GSE132042 | 32669714 | scRNA-Seq |
| Ext. Fig. e–i | Liver | *Mus musculus* | GSE132042 | 32669714 | scRNA-Seq |
| Ext. Fig. 6q | Bone | *Mus musculus* | GSE237301 | 37524694 | CyTOF |
| Ext. Fig. 7a–e | Spleen | *Mus musculus* | GSE132901 | 37972658 | scRNA-Seq |
| Ext. Fig. 7f–j | Kidney | *Mus musculus* | GSE132901 | 37972658 | scRNA-Seq |
| Ext. Fig. 8 | Liver | *Mus musculus* | GSE218300 | 37972658 | scRNA-Seq |
| Ext. Fig. 9 | MEF | *Mus musculus* | GSE117278 | 35764649 | bulk RNA-Seq |
| Ext. Fig. 10e–i | Skin | *Homo sapiens* | GSE130973 | 32327715 | scRNA-Seq |
| Ext. Fig. 11 | Lung | *Mus musculus* | GSE132901 | 37972658 | scRNA-Seq |
| Ext. Fig. 12 | Adipose tissue | *Mus musculus* | GSE269660 | 39111286 | scRNA-Seq |

Sections were deparaffinized in Histoclear (twice, 5 min each) and rehydrated through a graded ethanol series: 100% ethanol (twice, 5 min each), 90% ethanol (twice, 5 min each), 70% ethanol ($1 \times 5$ min), followed by two washes in distilled water (5 min each). Antigen retrieval was performed by heating the slides to boiling in citrate buffer (pH 6.0; Agilent-Dako, S236984) for 10 min. Slides were allowed to cool at room temperature for 30 min and then rinsed twice in PBS (5 min each).

Tissue sections were blocked with 1% BSA and 1:60 normal goat serum (NGS; Agilent-Dako, X090710-8) for 30 min at room temperature, followed by overnight incubation with primary antibodies at 4 °C. The next day, sections were washed in PBS and incubated with appropriate secondary antibodies for 1 h at room temperature. After three additional PBS washes, slides were mounted using ProLong™ Gold Antifade Mountant with DAPI (Invitrogen).

The primary antibodies used were anti-p16 (725-4793) mouse monoclonal (dilution as provided by kit, VENTANA) and anti-p21 (2947S) (1:200, Cell Signalling).

## MERFISH analysis

A punch biopsy was collected from the upper thigh of a 66-year-old individual. Following collection, the sample was cleaned and the surrounding adipose tissue was carefully removed. The tissue was then centered in a plastic embedding mold, embedded in OCT compound, and rapidly frozen by immersion into super-cooled isopentane, which had been pre-chilled in a metal container placed within a liquid nitrogen bath. Using forceps, the mold was carefully lowered into the isopentane–liquid nitrogen bath without full submersion. Once the OCT was fully solidified, the tissue blocks were stored at –80 °C until shipment to the manufacturer.

MERFISH was performed by the manufacturer (Vizgen) using the pre-designed ImmunOncology gene panel in combination with a custom-designed probe targeting *CDKN2A* (ENST00000579755). The ImmunOncology panel included 550 genes encompassing markers of cellular senescence, cell-cycle regulation, inflammation, immune cell types and others. All steps of the MERFISH workflow, including tissue preparation, hybridization, imaging, and data acquisition, were conducted by Vizgen according to their standardized protocols. The resulting data were exported using the MERSCOPE Visualizer software, version 2.3 (Vizgen, Inc.).

The spatial coordinates were extracted and transferred into a UMAP using the Seurat package (v 5.0.0).

## Statistics

Statistical analyses were performed using either GraphPad Prism (Version 9.0) or R version 4.2.0. A $p$ value $<0.05$ (two-tailed) was considered statistically significant.

## Ethics statement

Human sun-protected skin tissue samples were obtained from participants at Mayo Clinic and from the Baltimore Longitudinal Study on Aging (BLSA). At Mayo Clinic, skin biopsies were collected from consented participants under IRB protocol 21-008529, in accordance with the International Conference on

Harmonization, Good Clinical Practice guidelines, and the Declaration of Helsinki. Samples from BLSA participants were collected under a protocol approved by the Institutional Review Board of the Intramural Research Program of the National Institutes of Health (protocol #03AG0325). Human lung tissues were obtained incidentally during clinically indicated thoracic surgeries under IRB-approved protocols (Mayo IRB 08-002518), with informed consent obtained prior to collection. All samples were de-identified before analysis.

## Data availability

All datasets analyzed in this study were obtained from publicly available databases, except for the data underlying Extended Data Fig. EV10K–M. The raw data for this figure will be made available by the corresponding author upon reasonable request. The code for each figure is provided at: https://github.com/donshiva88/SaulLab_code/blob/main/P16_vs_P21.

The source data of this paper are collected in the following database record: biostudies:S-SCDT-10_1038-S44318-025-00601-2.

## Peer review information

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

## Acknowledgements

This research was supported by the NIH through grants P01 AG062413 (to SK, NKL, DM, DJ, LJN, PDR, and JFP), R01AG068048 (to JFP), R01AG82708 (to JFP), UG3CA26810308 and UH3CA268103 (to JFP), R01AG086085 (to SK and DM), Hevolution HR-GRO-23-1199144-8 (to SK), R01 AG068182 (to DJ), R01 AG063543 (LJN), Hevolution/AFAR (to DJ), R01 AG076515 (to SK, DM, PDR, LJN), U54 AG079754 (to SK, NKL, PDR, LJN), R01 AG055529 (to NKL), U54 AG076041 (to LJN and PDR), U19 AG056278 (to PDR and LJN), and NRF grant (MSIT, NRF-2020M3A9D8038014) (to YH and SYK).

## Author contributions

**Dominik Saul**: Conceptualization; Data curation; Software; Formal analysis; Validation; Investigation; Visualization; Methodology; Writing—original draft; Project administration; Writing—review and editing. **Diana Jurk**: Conceptualization; Resources; Data curation; Validation; Investigation; Writing—review and editing. **Madison L Doolittle**: Conceptualization; Data curation; Supervision; Validation; Investigation; Visualization; Writing—review and editing. **Robyn Laura Kosinsky**: Conceptualization; Validation; Investigation; Visualization; Writing—review and editing. **Yeaeun Han**: Data curation; Formal analysis; Validation; Investigation; Visualization; Project administration; Writing—review and editing. **Xu Zhang**: Data curation; Methodology; Writing—review and editing. **Ana Catarina Franco**: Data curation; Formal analysis; Validation; Investigation; Visualization; Methodology; Writing—review and editing. **Sung, Y Kim**: Project administration; Writing—review and editing. **Saranya P Wyles**: Data curation; Supervision; Validation; Writing—review and editing. **Y S Prakash**: Data curation; Investigation; Writing—review and editing. **David G Monroe**: Data curation; Methodology; Writing—review and editing. **Luigi Ferrucci**: Data curation; Writing—review and editing. **Nathan K LeBrasseur**: Resources; Data curation; Supervision; Funding acquisition; Investigation; Project administration; Writing—review and editing. **Paul D Robbins**: Data curation; Supervision; Validation; Writing—review and editing. **Laura J Niedernhofer**: Supervision; Validation; Investigation; Writing—review and editing. **Sundeep Khosla**: Conceptualization; Resources; Data curation; Software; Supervision; Funding acquisition; Validation; Investigation; Visualization; Methodology; Writing—original draft; Project administration; Writing—review and editing. **João, F Passos**: Conceptualization; Resources; Data curation; Software; Supervision; Funding acquisition; Validation; Investigation; Visualization; Methodology; Writing—original draft; Project administration; Writing—review and editing.

Source data underlying figure panels in this paper may have individual authorship assigned. Where available, figure panel/source data authorship is listed in the following database record: biostudies:S-SCDT-10_1038-S44318-025-00601-2.

## Disclosure and competing interests statement

The authors declare no competing interests.

# Expanded View Figures

**Figure EV1.  Comprehensive examination of the mRNA expression profiles for the whole secretome and SenMayo in aging murine hippocampus.**

(A) *p16+* cells are significantly more frequent in the old brain. (B) Microglia has more *p21+* cells in the old brain. (C) In the murine hippocampus, there is minimal overlap between the mRNA expression of the whole secretome + SenMayo in *p21+* and *p16+* cells. (D) SenMayo composition in *p16+* vs. *p21+* microglia shows some common secretory genes like *Plaur* and *Cxcl16*, while in (E) oligodendrocytes, there is no common secretory phenotype. (F) The velocity trajectory of *p16+* cells, *p21+* cells and the remaining brain cells shows no overlap or common origin (G) Monocle3 applied to the backbone from f shows that there is no common "ancestor" for these cell types. **A**: unpaired *t*-test, $n = 2$ per condition, error bars: sd, **B**: multiple paired *t*-tests, $n = 2$ per condition, error bars: sd.

▶

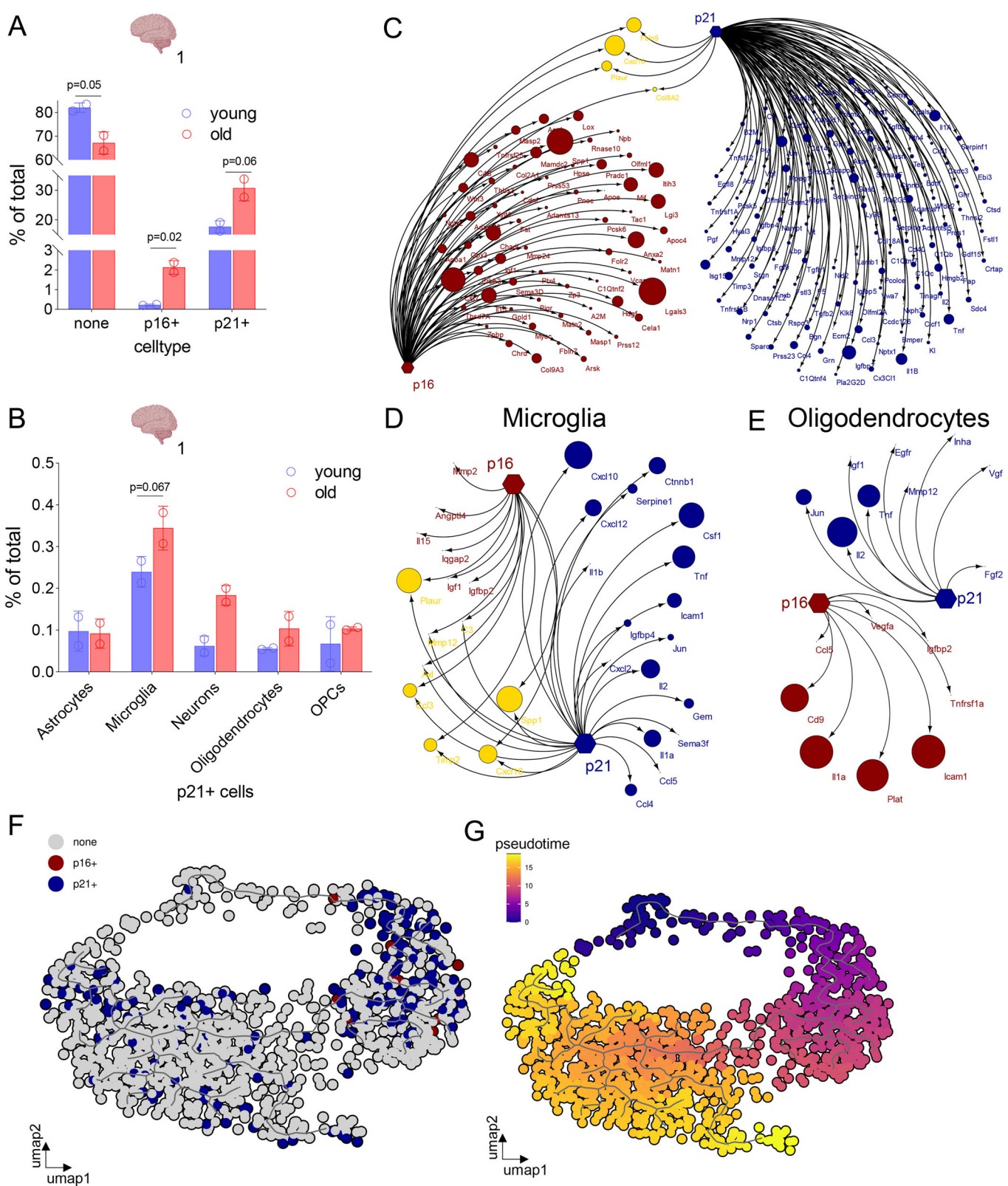

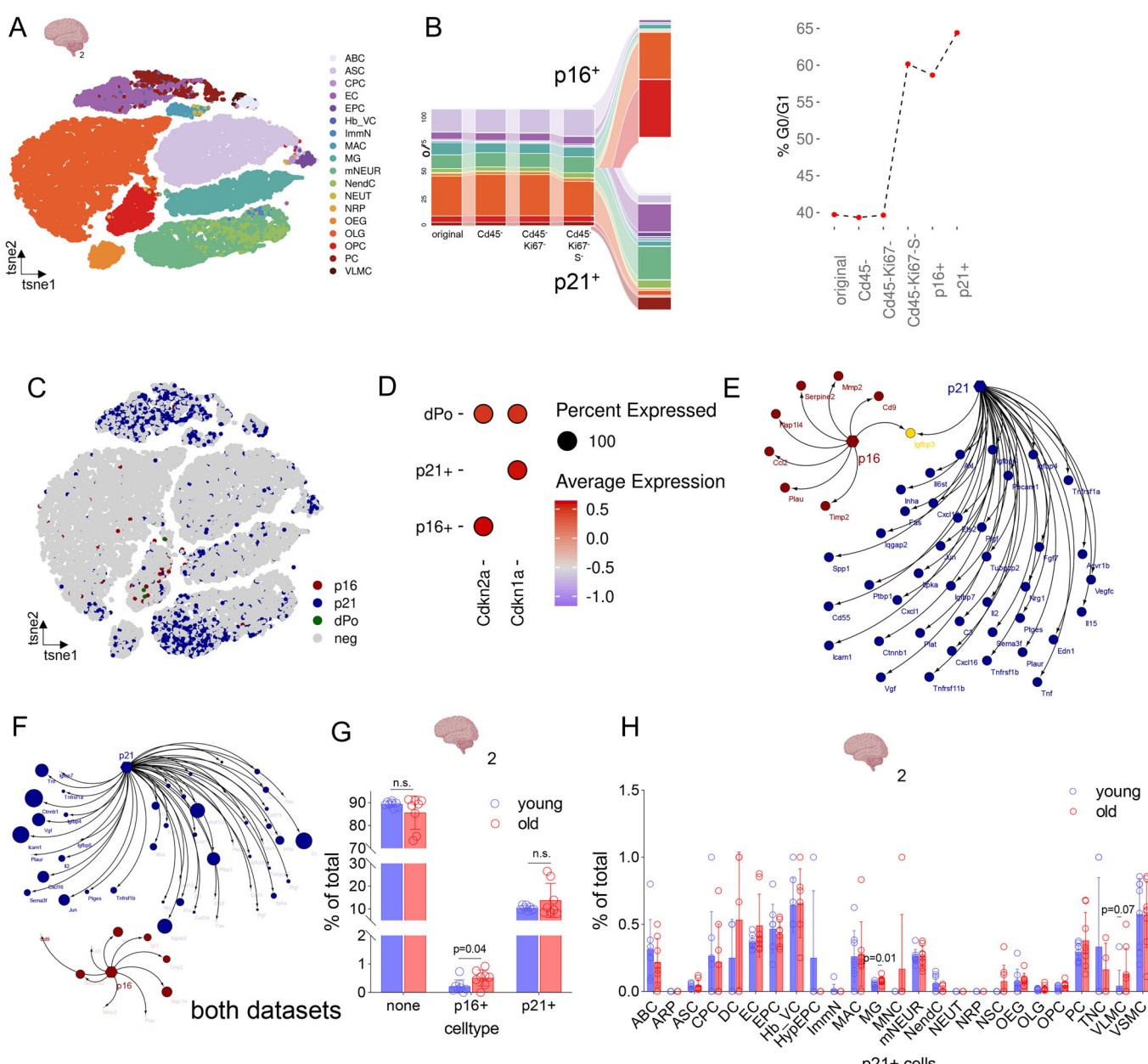

**Figure EV2. A second brain sample (Ximerakis et al, 2019) was used to confirm the findings from the first brain sample.**

(A) A t-SNE plot indicating all cell types. (B) The selection of *p16+* cells focuses on oligodendrocyte precursors, while *p21+* cells were mostly endothelial cells. (C) The t-SNE plot demonstrates that most cells are either *p16+* or *p21+*, but double-positive cells are rare. (D) The expression of *p16+*, *p21+* cells, and the rare dPo population. (E) There is only a very small overlap between the *p16-* and *p21-*positive secretory phenotype, namely just one gene (*Igfbp3*). (F) When combining both brain datasets, we observed no consistent overlap between the secretory profiles of *p16+* and *p21+* cells, underscoring that they represent distinct populations. Shared genes across both datasets are shown in solid color, whereas genes unique to a single dataset are displayed transparently. (G) *p16+* cells are significantly more frequent in the old brain. (H) Microglia (MG) has more *p21+* cells in the old brain. (G): unpaired *t*-test, *n* = 8 per condition, error bars: sd, (H) multiple paired *t*-tests, *n* = 8 per condition, error bars: sd.

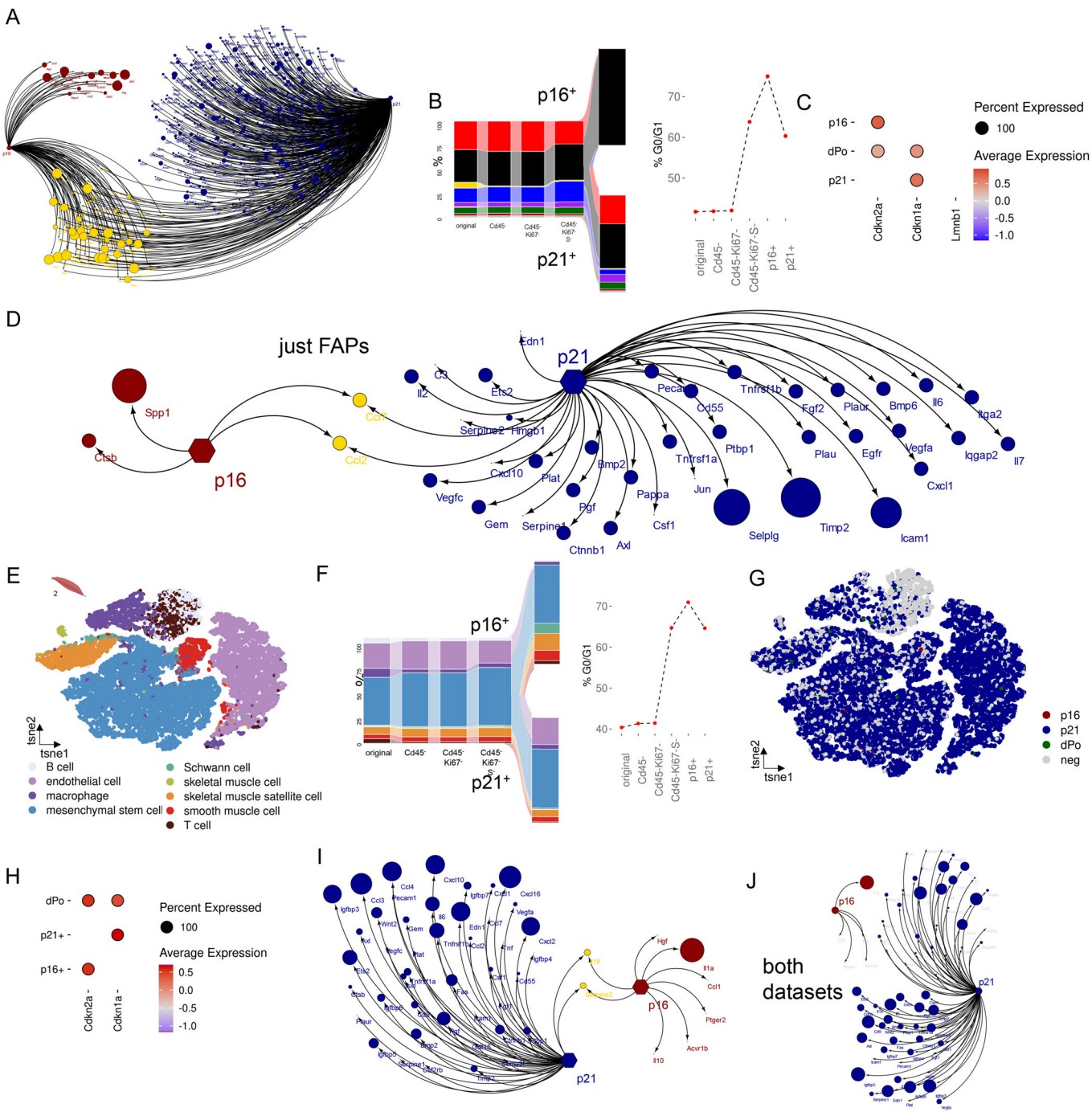

**Figure EV3.  Comprehensive analysis of mRNA expression profiles in aging murine muscle.**

(**A**) In the muscle, there is some overlap with a noticeably larger p21-associated secretome compared to p16. (**B**) The selection criteria for *p21+* and *p16+* cells. While most of the *p16+* cells are FAPs (black), in the *p21+* cells there are also some endothelial cells (red). (**C**) Key markers for the abovementioned populations. (**D**) Secretory phenotype composition in *p16+* and *p21+* FAPs. (**E**) A second muscle dataset (Tabula_Muris_Consortium, 2020) showed nine key cell populations, with (**F**) most of the *p16+* and *p21+* cells originating from the mesenchymal stem cell compartment. (**G**) The majority of cells is *p21+*. (**H**) Key markers for these three populations. (**I**) The secreted factors between these two populations are heterogeneous, and (**J**) Integration of both datasets shows no overlap in secreted factors between these two populations.

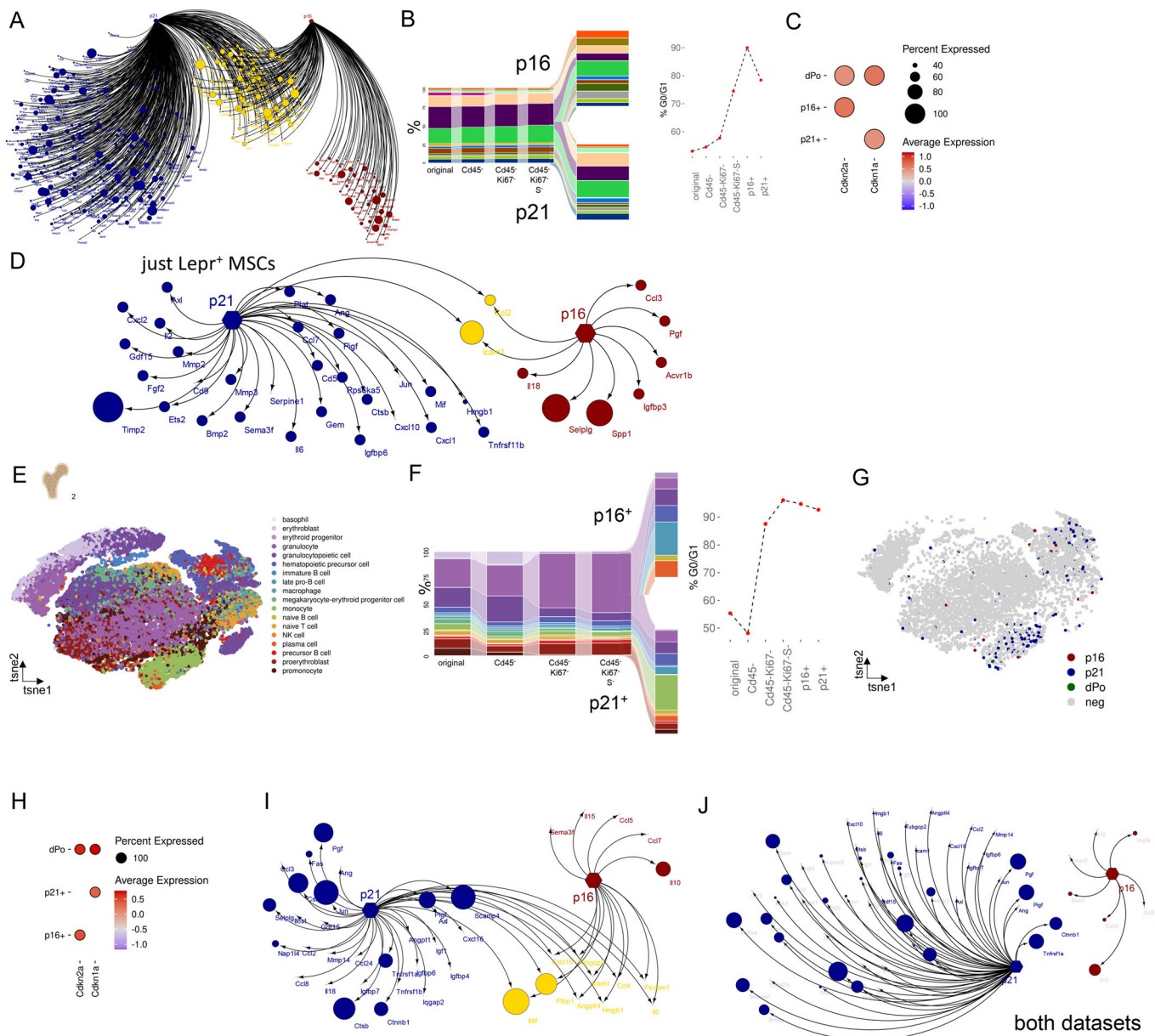

**Figure EV4. Analysis of the secretome in bone during aging.**

(A) Similar to the muscle, in the bone, there is some overlap but a more pronounced p21-associated secretome. (B) The selection criteria and key populations (C) for *p16+* and *p21+* cells. (D) Just Lepr+ MSCs and p16- and p21-associated SenMayo phenotype, respectively. (E) A second bone dataset (Tabula_Muris_Consortium, 2020) shows a (F) high amount of B cells in *p16+* cells and monocytes in *p21+* cells. (G, H) There are a few *p21+* cells, and almost no dPo cells. (I) There is some overlap between *p16+* and *p21+* cells, while a summary of both bone datasets (J) shows no single gene overlapping for *p16-* and *p21-*associated secretory factors.

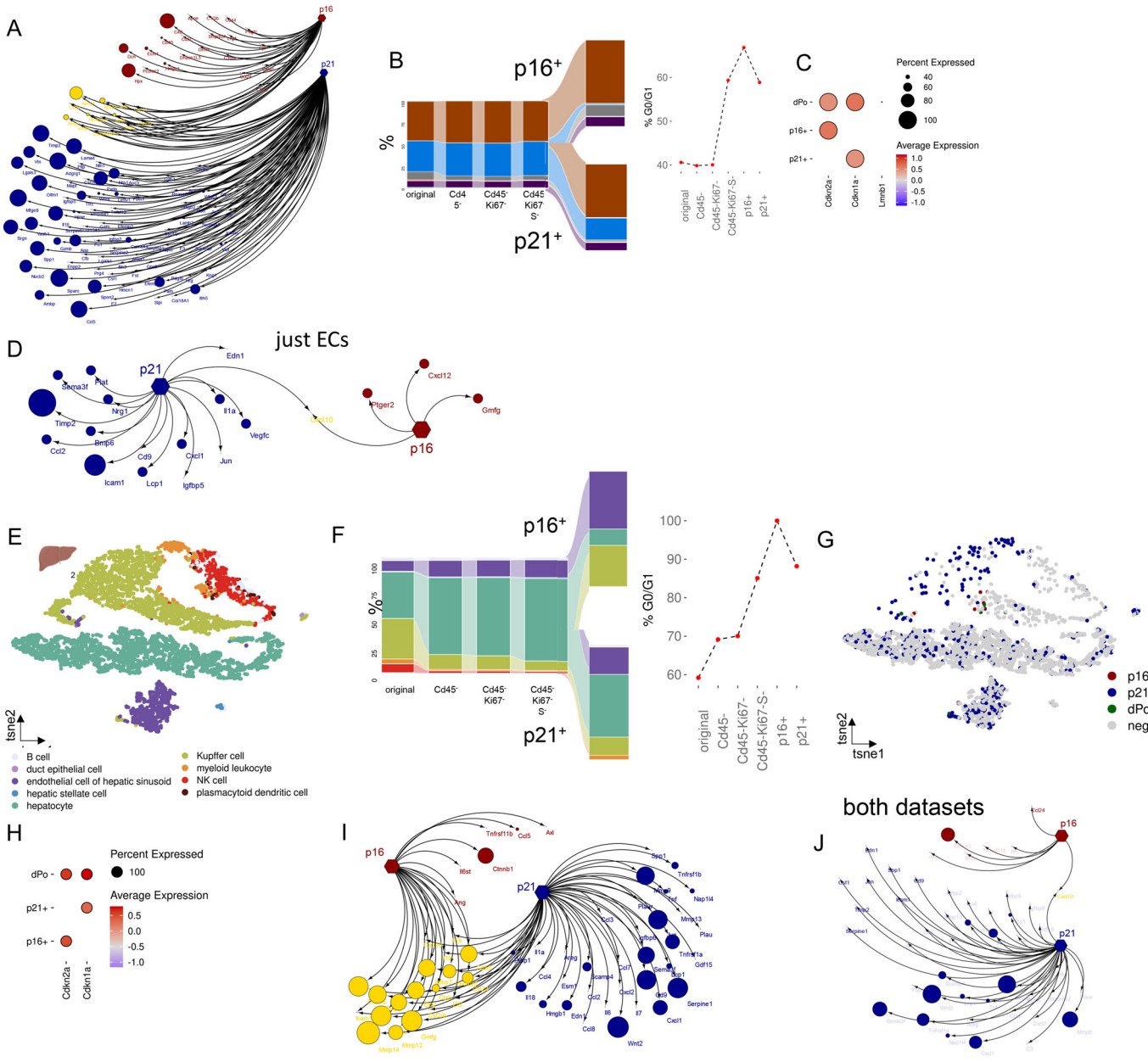

**Figure EV5. Liver analysis demonstrates differences between *p21*+ cells and *p16*+ cells in aging.**

(**A**) Within the liver, there is minimal overlap between *p21*+ cells and *p16*+ cells, with the *p21*+ cells predominantly expressing the majority of factors. (**B**) The selection of *p16*+ cells shows mostly endothelial cells, while in the *p21*+ cells, there are also some hepatocytes. (**C**) The key markers for the three key populations. (**D**) *p21*- and *p16*-associated SenMayo secretory phenotype in ECs. (**E**) A second hepatic dataset (Tabula_Muris_Consortium, 2020) was analyzed, while (**F**) endothelial cells were the most abundant in *p16*+ cells, and hepatocytes the most prominent in the *p21*+ cells. (**G**, **H**) *p21*+ cells were the majority compared to *p16*+ cells, and (**I**) there was a substantial overlap between *p21*- and *p16*-associated secretory factors, although *p21*+ cells expressed a greater number of factors than *p16*+ cells. (**J**) Combining both datasets, there was just one overlapping factor (*Cxcl16*).

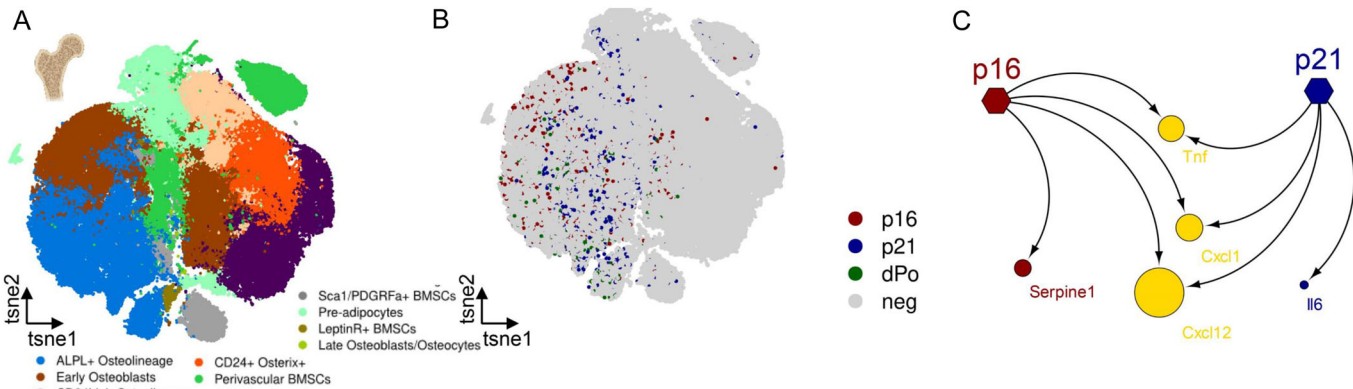

**Figure EV6. CyTOF analyses of *p16, p21* and associated secreted factors in murine bone.**

(A) CyTOF dataset from murine bone (Doolittle et al, 2023). (B) *p16+* and *p21+* cells are largely distinct, with minimal overlap. (C) The associated secretory phenotype of *p16+* and *p21+* cells shows some overlap but is limited to only a few factors, likely reflecting the limited antibody panel available.

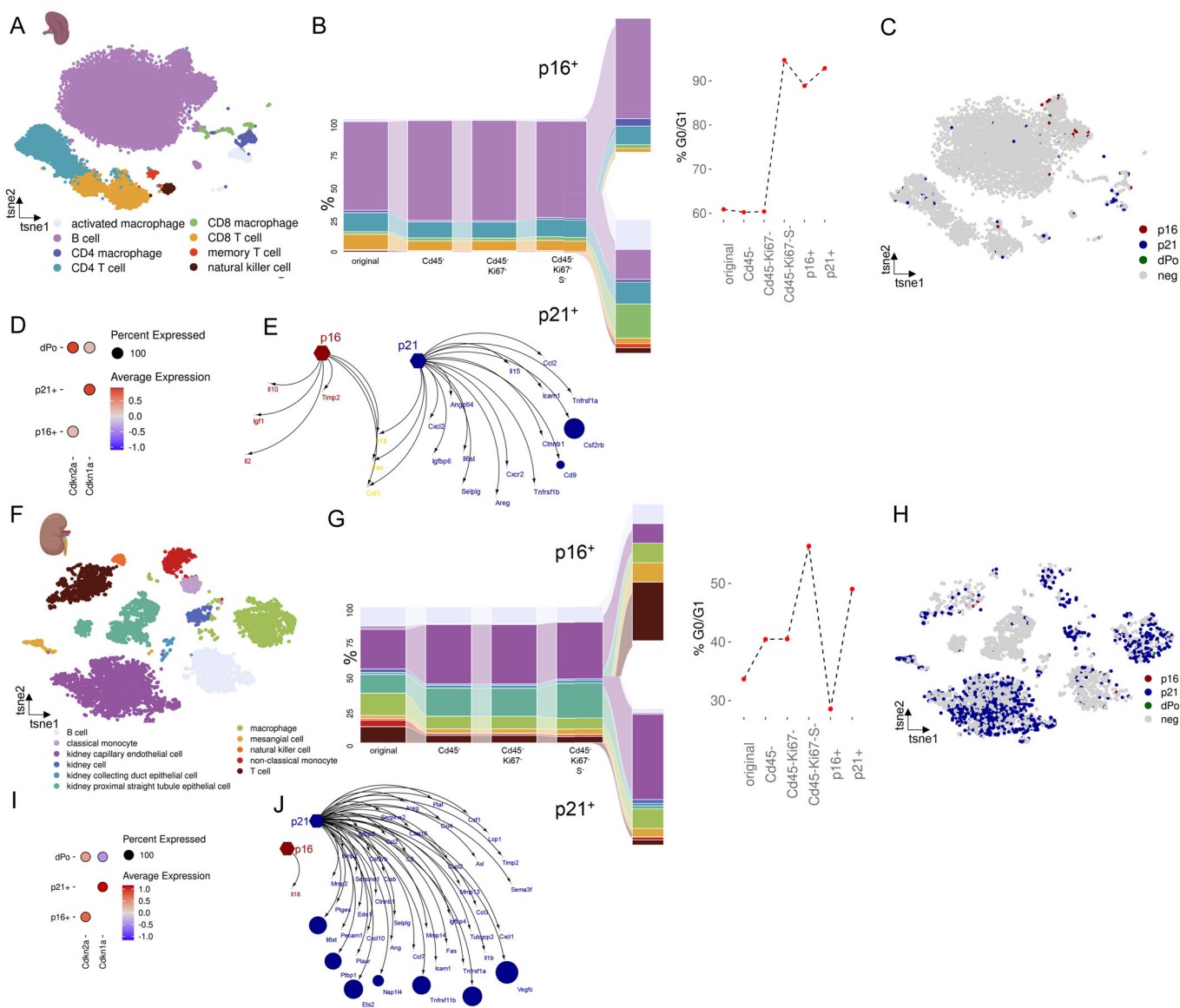

**Figure EV7.  Spleen and kidney display p16- and p21-associated differences consistent with previous murine datasets.**

(**A**) In the spleen calico dataset (Kimmel et al, 2019, GSE 132901), eight different populations were identified and (**B**) analyzed according to their Cd45-negativity, Ki67-negativity and S-phase exclusion, followed by *p16*- (mostly B cells) and *p21*-positivity (mostly Cd8 macrophages). (**C**) The majority of cells were *p21*+, although (**D**) only a small fraction of total cells were positive for either marker. (**E**) Secreted factors showed limited overlap, with most associated with *p21*+ cells. (**F**) The kidney dataset consisted of 11 different cell types, out of which (**G**) T cells were the most abundant in *p16*+ cells and capillary endothelial cells were the most abundant in *p21*+ cells. (**H, I**) The majority of cells was *p21*+. (**j**) A vast majority of secreted factors was associated with *p21*+ cells.

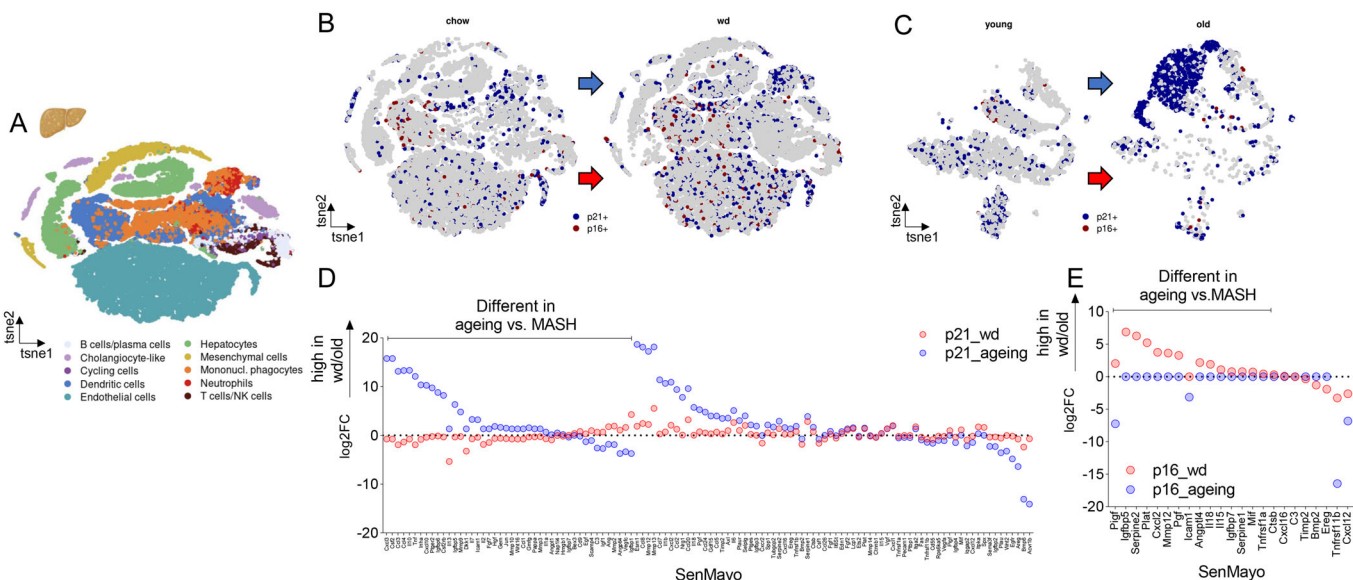

**Figure EV8. Distinct secretory phenotypes of *p16*+ and *p21*+ cells in MASH and aging liver.**

(A) Analysis of a diseased liver (MASH, GSE218300, (Bendixen et al, 2024)), (B) a western-style diet (wd) compared to chow resulted in a moderate increase of *p21*+ cells, while (C) aging resulted in a profound increase of *p21*+ cells. (D) The secretory phenotype in *p21*+ cells was different to a large degree in aging vs. MASH, with a higher amount of genes upregulated in aging. Similarly, the (E) secretory phenotype in *p16*+ cells was markedly different in aging vs. MASH, but with more genes upregulated in the western diet compared to aging.

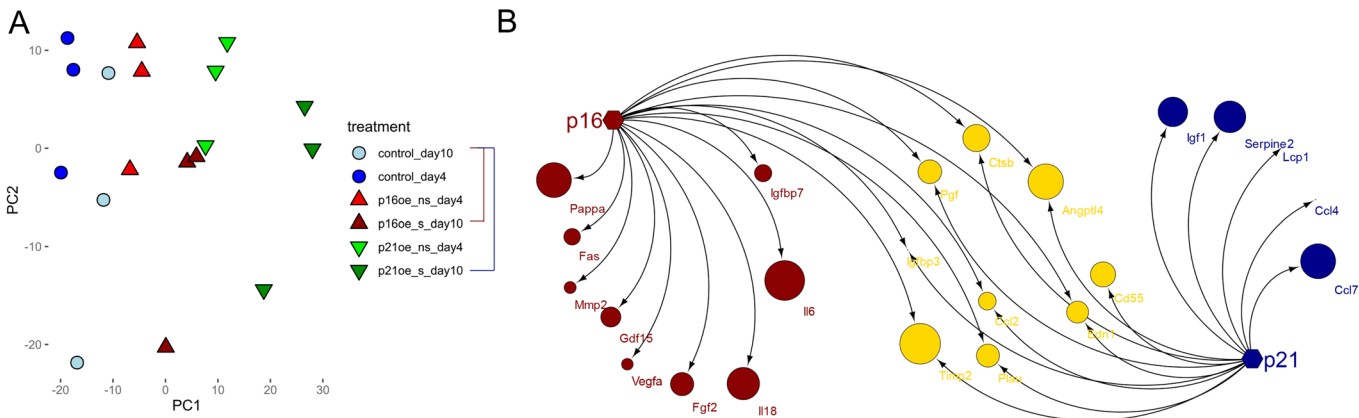

**Figure EV9.  Overexpression of p16 and p21 reveals distinct secretory profiles (GSE117278).**

(A) A PCA plot illustrating control-MEFs on day 4 and 10, as well as p16-overexpressing cells (adeno-Cre-EGFP virus Ai14;L-p16 injection into the tail) on day 4 and 10, compared to p21-overexpressing cells (Ai14;L-p21) on day 4 and 10. (B) A comparable number of factors are expressed in p16+ (red)- vs. p21+ (blue) cells, with overlap indicated in yellow.

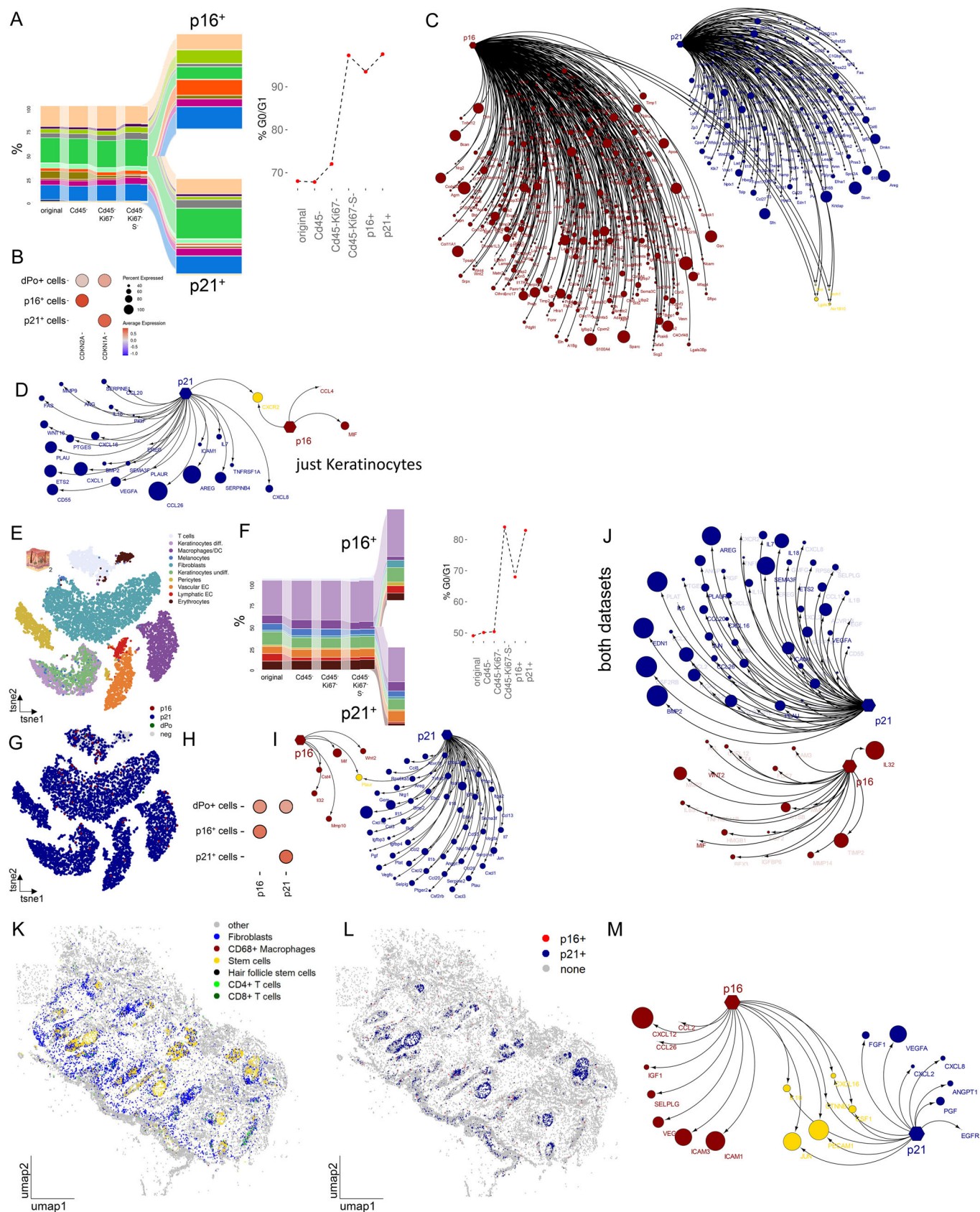

◀ **Figure EV10. Human skin secretome, associated with p21 and p16, incorporating datasets from Zou et al, (2021) and Solé-Boldo et al, (2020).**

(A) In the skin, there is minimal overlap between *p16+* and *p21+* cells in terms of mRNA expression of the whole secretome. Unlike the majority of examined tissues, the mRNA expression of the entire secretome associated with p16 is as extensive as that associated with p21. (B) CD45-negativity, followed by Ki67-negativity and exclusion of S-phase cells were followed by p21- or p16-positivity for selection of proposedly senescent cells, which key markers are depicted in (C). (D) SenMayo in *p16+* and *p21+* keratinocytes, respectively. (E) Five human skin samples (GSE130973) were likewise (F) applied these selection criteria, but *p16+* cells were lacking, resulting in an abundance of (G) *p21+* cells and (H) no *p16+* cells. (I) Subsequently, there is just a p21-associated SASP, and no overlap (J) in both datasets for *p16+* and *p21+* cells. (K) Multiplexed error-robust fluorescence in situ hybridization (MERFISH) analysis on human skin samples, with different cell types within a human skin sample. Within these cell types, (L) *p16+* (red) and *p21+* (blue) cells are largely distinct, with minimal overlap between these markers. (M) *p16+* and *p21+*-associated secretory profiles within the MERFISH dataset.

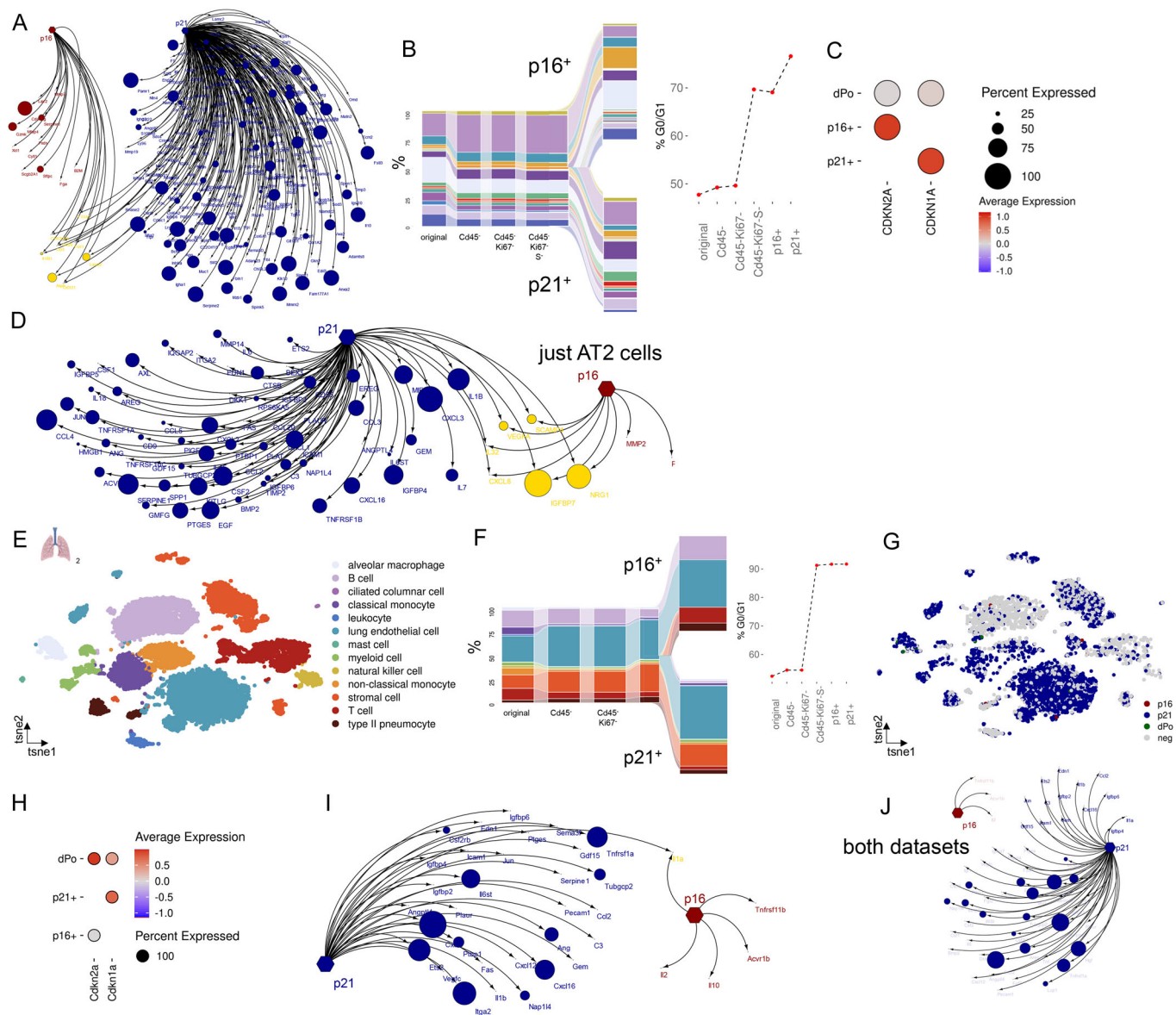

**Figure EV11. Analysis of lung reveals limited overlap in the secretory phenotype and distinct developmental profiles of p16- and p21-positive cells.**

(A) In the human lung, the *p21+* cells express a large number of secreted factors, with a very small overlap with *p16+* cells. (B) Selection criteria for human lung tissue include CD45-negativity, followed by Ki67-negativity and exclusion of S-phase cells, before the cells are distributed between p16+ or p21+. (C) The markers for these three key populations. (D) *p16-* and *p21-*SenMayo associated secretory phenotype in *p16+* and *p21+* AT2 cells, respectively. (E) The murine Calico lung dataset (Kimmel et al, 2019) provides 13 different cell types, which are selected (F) as abovementioned. (G, H) The vast majority of cells are *p21+*. (I) The secretory phenotype itself is more abundant in the *p21+* cells and there is (I) a small overlap of *p16+* and *p21+* associated factors between both datasets.

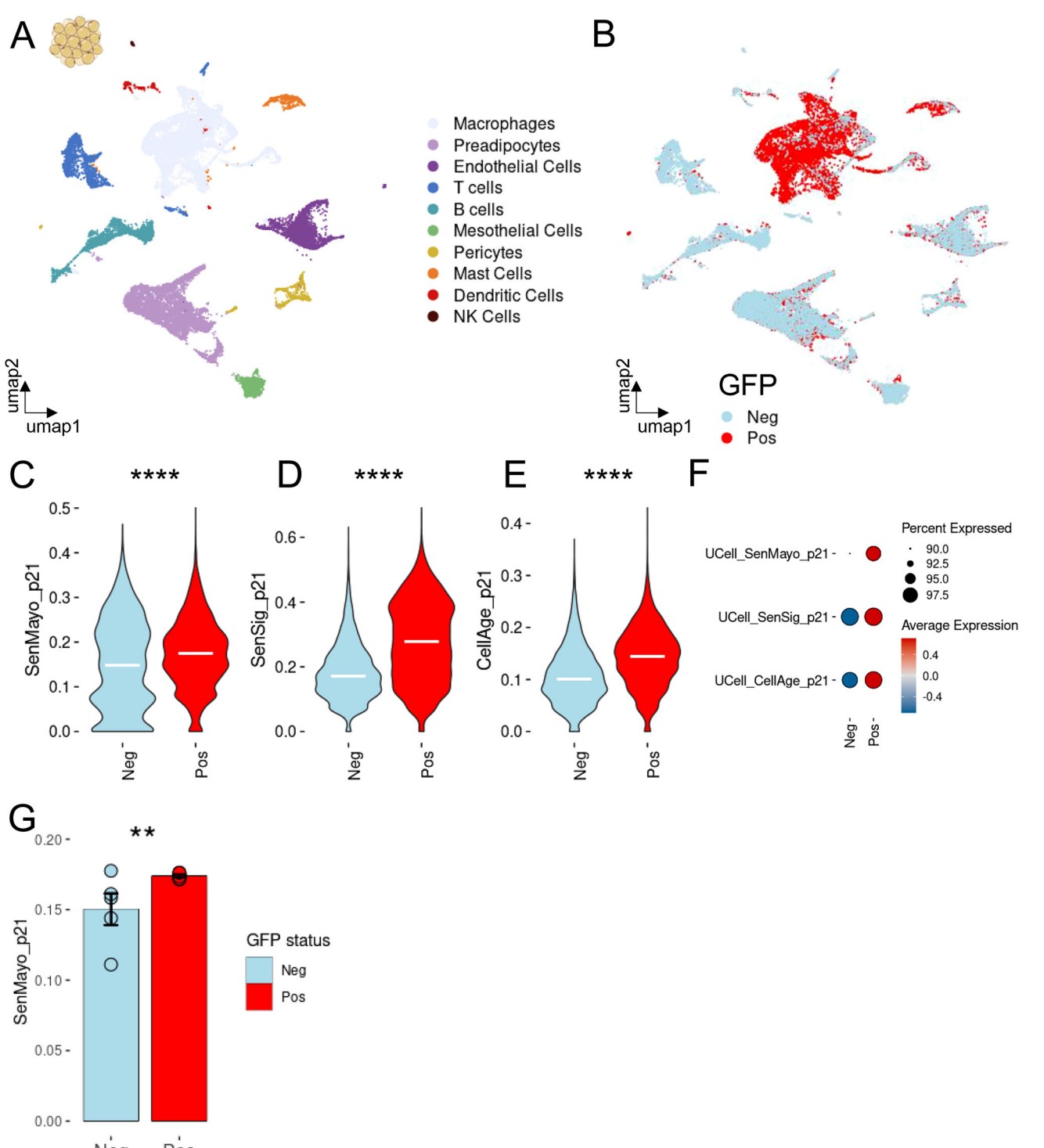

**Figure EV12.   In an adipose tissue dataset (Wang et al, 2024a, GSE269660), the GFP/Cdkn1a-positive cells exhibited an enrichment of the p21-associated secretory phenotype.**

(**A**) In adipose tissue, ten different populations were identified, out of which the (**B**) macrophages were the highest in *p21/GFP*. (**C**) The *p21*-associated SenMayo, (**D**) SenSig, and (**E**) CellAge were all (**F**) significantly higher in the *GFP*-positive cells compared to the *GFP*-negative cells. (**G**) Likewise, a samplewise comparison ($n = 10$) between *GFP*-positive ($n = 5$) and *GFP*-negative ($n = 5$) samples shows an increase of *p21*-associated SenMayo in the positive samples. ****$p$ value <0.0001, ***$p$ value <0.001, **$p$ value <0.01, *$p$ value <0.05. (**C–E**) Exact $p$ values = $2 \times 10^{-16}$, Wilcoxon test, (**G**) Exact $p$ value = 0.004, *t*-test, error bars: 2 SE. $n = 5$ per condition

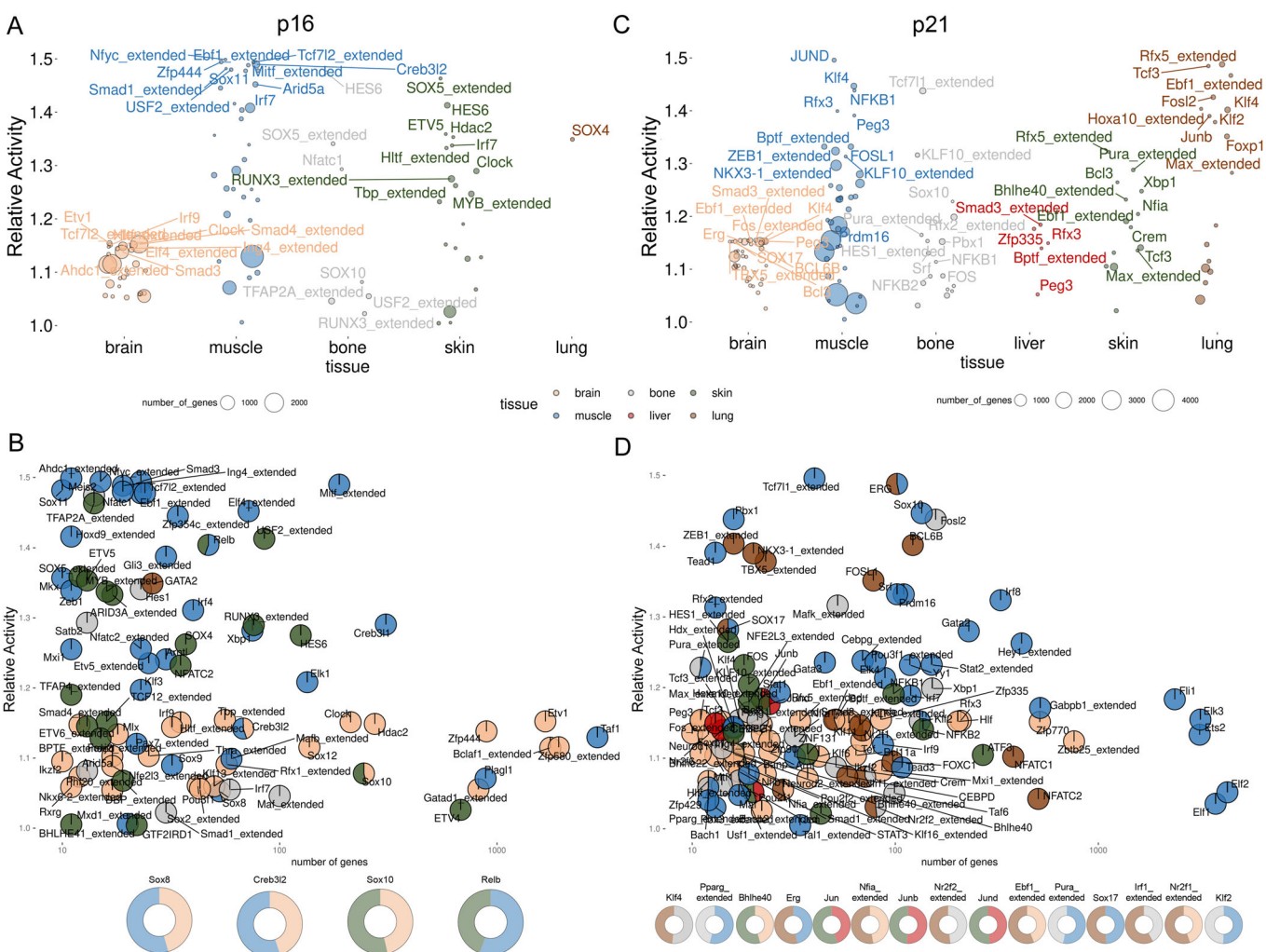

**Figure EV13. SCENIC analysis of the regulating transcription factors of *p16+* and *p21+* cells.**

(A) When analyzing five distinct tissues for regulatory factors in *p16+* cells, each tissue exhibits a substantial array of transcription factors governing the behavior of *p16+* cells. (Since the liver has very few p16-positive cells for proper calculation with SCENIC, these were excluded). (B) The x-axis represents the number of genes, plotted against the y-axis, illustrating the relative activity of the respective transcription factor, highlighting the significance of each factor. Interestingly, only four factors (*Sox8, Creb3l2, Sox10,* and *Relb*) exhibit consistency across multiple tissues in the context of *p16+* cells. (C) In *p21+* cells, the regulating transcription factors show a high heterogeneity across tissues. (D) The transcription factors associated with *p21* are predominantly specific to a single organ, but a few transcription factors, such as *Erg, Sox17, Klf4, Jun, Klf2,* and others, regulate *p21+* cells in two tissues.

