## [Peer Review File · The EMBO Journal]

Distinct senotypes in p16- and p21-positive cells across human and mouse aging tissues

Dominik Saul, Diana Jurk, Madison Doolittle, Robyn Kosinsky, Yeaun Han, Xu Zhang, Ana Catarina Franco, Sung Kim, Saranya Wyles, Y Prakash, David Monroe, Luigi Ferrucci, Nathan LeBrasseur, Paul Robbins, Laura Niedernhofer, Sundeep Khosla, and Joao Passos

Corresponding authors: Joao Passos (Passos.Joao@mayo.edu) , Sundeep Khosla (khosla.sundeep@mayo.edu)

Review Timeline:

Submission Date: Editorial	8th Jul 25
Decision:	13th Aug 25
Editor's Correspondence:	30th Aug 25
Revision Received:	23rd Sep 25
Accepted:	30th Sep 25

Editor: Daniel Klimmeck

Transaction Report:

This manuscript was reviewed by another journal and then transferred to an EMBO Press journal. EMBO Press has a transfer agreement transparent review process policy.

Reviewers' Comments (first submission):**Reviewer #1:**

Remarks to the Author:

In this manuscript, Passos and colleagues have performed in-depth, single-cell transcriptomic analysis of different organs to identify senescent cells. They have focused on those cells that selectively express p21 mRNA, p16 mRNA, or both p21 and p16 mRNAs. After finding that senescent cells largely express one or the other cdk inhibitor mRNA, they identify different mRNAs encoding SASP factors present in cells that also express p21 mRNA, p16 mRNA or both mRNAs.

The authors are correct in saying that the field needs more detailed knowledge of the unique features of senescent cells. There is already widespread appreciation in the field that senescent cells are exceedingly heterogeneous, depending on cell type, stressor type and intensity, time after senescence induction, metabolic state, etc. In fact, the authors themselves have led efforts to understand this heterogeneity. Unfortunately, I am afraid the current study does not advance the field substantially, with some of the key problems identified below. Perhaps most importantly, at the present time, a manuscript that totally relies on bioinformatic analysis without validation/follow-up biology is more appropriate for a bioinformatic journal.

Major comments:

There are already numerous single-cell transcriptomic analyses of senescent cells from organs that highlight the immense heterogeneity the authors underscore here. In fact, some of those very studies are the starting point for parts of the current manuscript. This reliance on published datasets is also partly a concern: the authors do not generate much new data, and instead reanalyze earlier datasets. I appreciate the value in reanalyzing published datasets, but it does reduce the novelty of this study.

I was (briefly) excited to see that the authors attempted a limited parallel analysis of the transcriptomes by studying some proteins (CyTOF). Unfortunately, this analysis was too limited and preliminary to really contribute a valuable proteomic dimension to the study.

My enthusiasm is probably most curbed by the fact that all the entirety of the analysis is bioinformatic. The authors have not included biological validation of their computational results/predictions. If this validation had been done by *in situ* analysis of the mRNAs that are predicted to be coexpressed in a given cell (mRNAs encoding p21, p16, p21+p16 and various SASP factors) or better yet, the proteins themselves, the work would have been truly enriched. Single-cell validation using flow or other single-cell methods would also have gone a long way towards strengthening the authors' hypotheses with relevant biology.

Finally, two minor but important points could be added in regard to the assembly of the manuscript:

- One is that the writing is quite 'loose', some of it reflecting a worrisome trend among data scientists to use 'gene' to refer to 'mRNA' and sometimes also to 'protein'. Here, the authors have limited their analysis to 'p21 mRNA' and 'p16 mRNA'—yet they consistently write instead 'p21' and 'p16' (the proteins). It is not clear if they are simply using incorrect nomenclature, or they are assuming that a cell that expresses p21 mRNA also expresses p21 protein. Of course, both points need attention and correction.

- The other is that the graphs look pretty cool, but they are not at all intuitive or easily interpretable. I would urge the authors to use more mainstream representation that readers can follow without massive difficulty. This concern also reflects my earlier recommendation that this manuscript be published in a journal with a more bioinformatic audience.

Reviewer #2:

Remarks to the Author:

The manuscript written by Dominik Saul et al. investigated p16- and p21-expressing cells in the single-cell RNA-seq datasets of multiple organs from mice and humans. The authors identified distinct secretomes associated with p16 and p21, while identifying factors common to both datasets ("core" SASP and "common" SASP). They also performed CellChat and SCENIC analyses to understand the functional significance of the transcriptomic differences. This study contains a substantial amount of data, some of which provide novel insights, but there are several fundamental questions that the authors need to address:

Revision points

a) Major

1) Definition of senescent cells: The authors identified Ki-67-negative S-negative p16- and/or p21-positive cells as senescent cells in the scRNA-seq datasets. However, as the authors state in the introduction, p16 and p21 expression is not restricted to senescent cells. The authors have not confirmed that these cells are in a state of cell cycle arrest, or that these cells increase with age. Selection of cells by cell cycle stage and Ki-67 expression may improve identification, but is not sufficient to claim that a significant proportion of these cells are senescent. In particular, p21 is expressed in various non-senescent cells/states, which somewhat explains the authors' observation of p21 expression in many cells. Therefore, the authors should be more careful when using the terms "senescent" or "SASP" in the manuscript, including the manuscript title. For example, I suggest replacing "senescent cells" with "p16/p21-expressing cells" or "p16/p21-expressing non-proliferating cells", and "SASP" should be replaced with "p16/p21-associated secretory phenotype" or "p16/p21-associated secretome", when referring to the observation in the scRNA-seq datasets.

2) Novelty: The main findings of this study are that p16- and p21-associated secretomes are distinct *in vivo*, and that the secretomes are highly diverse across cell types and organs. However, as the authors note in the manuscript, it is already known that p16 and p21 are associated with distinct secretomes and biologies *in vitro*. This study may be the first to characterize this distinction *in vivo*, but I am afraid the distinction is something that everyone would expect. The biological/functional heterogeneity of senescent cells in different organs/cell types has also been already reported in several papers. Some of the factors discussed by the authors may not have been previously reported, but the manuscript focuses solely on describing each factor, rather than interpreting the results from a more general perspective. Therefore, more in-depth analyses and previously unknown findings are needed to claim novelty. For example, is the difference between p16- and p21-associated secretomes parallel *in vivo* and *in vitro* (Sturmlechner I et al. 2021. Science)? Do trajectory analyses show results consistent

with the "p21 in the early phase and p16 in the late phase" model (Stein GH et al. 1999. Molecular and Cellular Biology), and how do the secretory factors change at the transition? How are the results of the CellChat/SCENIC analyses consistent (or not) with previous reports? How are p16/p21-related phenotypes affected by interactions with other cells and environmental cues?

3) Confirmation of the findings: I understand that the strength of this study is the observation of the heterogeneity of the potentially senescent cells in vivo. However, almost all of the results in this paper are based on single-cell RNA-seq data. As the authors may be aware, single-cell RNA-seq data are prone to multiple biases, such as cross-contamination between cell types and dropouts. Combining multiple datasets is not sufficient to guarantee that these biases are absent. Also, the difference in results between different organs may simply reflect the difference between experimental procedures, as the data from different organs came from different experiments. The CyTOF results do not seem to support the authors' claims as written in minor point (1). Furthermore, if the authors are to use terms such as "core SASP", or say that the p16- and p21-associated secretory phenotypes are distinct in vivo, it is important to confirm that the secreted factors are differentially secreted, or at least show that they are differentially expressed at the protein level in vivo. Immunostaining for p16, p21, and the major putative SASP factors in multiple organs, and checking whether the results parallel with the scRNA-seq findings should be adequate to address these issues. If the authors are unable to provide such data based on other experimental methods, it should be better to use terms such as "core p16/p21-associated secretory genes" or "core putative p16/p21-associated secretome" instead of "core SASP", and to use less assertive language regarding the difference in secretomes.

Minor

1) CyTOF: In the CyTOF analysis, three out of five factors are associated with both p16 and p21 (Fig. 2l). This seems to contradict the authors' claim that the p16- and p21-associated secretomes are highly distinct. In fact, most of the secretome factors are not shared between p16 and p21 in the bone scRNA-seq datasets (Fig. 2f, Extended Fig. 2b). As the only protein-level analysis in the study shows a result contrary to the story of the paper, the authors need to address what to make of this result. The small number of targets in CyTOF does not explain this discrepancy (the p16-p21 overlap percentage should not change if the target gene set is simply narrow down without bias). Is the difference between p16 and p21 not so pronounced at the protein level? Are the antibody targets in the CyTOF dataset biased? Or are the factors shared by p16- and p21-associated secretomes more easily captured on CyTOF? If either of these seems to be the case, what is causing this bias?

2) Further details on cell selection: The authors have shown the proportions of cell types in the course of selecting Ki-67-negative, S-negative, p16- and/or p21-positive cells (Fig. 1b). If they could show similar figures for all the datasets in the Extended Figures, the readers would be able to better understand how the cells were narrowed down. It would also be important to write what percentage of the cells in the parent group were selected in each selection step. In addition, related to the main point (1), looking at the Ki67+% and predicted cell cycle stages in CD45- p16/p21+ cells might be helpful to speculate what proportion of senescent cells are contained in the p16/p21-expressing cells.

Reviewer #3:

Remarks to the Author:

In this work, the authors employ public scRNA-seq datasets to compare the secretomes of p16- and p21-positive senescent cells. This is a timely and important topic, the paper is well-written, and the results are interesting, yet I also feel the analysis is somewhat preliminary and more could be done, as detailed below.

My major concern is that the authors call theirs a "comprehensive analysis", including in the abstract, yet it doesn't feel that way because several major datasets are not included and the work leaves several questions open. In particular, the authors don't include the larger scRNA-seq datasets of aging I'm aware of, namely the Tabula Muris Senis and the Calico murine aging cell atlas:

<https://pubmed.ncbi.nlm.nih.gov/32669714/>

<https://pubmed.ncbi.nlm.nih.gov/31754020/>

Related to the above, I couldn't find information on why certain data sets were selected, but not others. I think either the authors need to perform a comprehensive analysis incorporating at least the major publicly available scRNA-seq datasets or explain their criteria for data selection.

The main results are that there are substantial differences between the secretomes of p16- and p21-positive senescent cells. I think this is an interesting observation, but it also leaves a few open questions. In particular, I couldn't find results on the number or frequency of p16- and p21-positive senescent cells, and how these change with age. The original analysis performed on the Calico data found very few cells expressing markers of senescence and no significant changes in aged tissues, so determining whether the percentage of cells exhibiting markers of senescence changes with age would be important.

The authors limit their analysis to the secretome. I wonder why not perform a broader analysis of gene expression changes between p16- and p21-positive senescent cells?

In addition, I think it would be good to validate at least some results in another data set of senescence signatures, like for example our CellAge database:
<https://genomics.senescence.info/cells/>

Also, why limit the analysis to p16 and p21 as markers of senescence? For instance, a recent study employed a transfer learning for signatures to identify senescent cells from single cell data:

<https://link.springer.com/article/10.1007/s11357-023-00785-7>

A couple of minor comments:

Line 112: "when we analyzed an extensive list of secreted proteins not limited to the known SASP14 plus SenMayo" - what does this mean exactly? How many secreted proteins were analyzed?

Line 338: why not make the code available on GitHub?

Lastly, I think the work could be better placed in the context of previous findings. As above-mentioned, for instance, the Calico analysis studied senescent cells from single cell data. Our analysis of intercellular communication in aging using scRNA-seq data also revealed known SASP factors associated with the increase in inflammation with age:
<https://pubmed.ncbi.nlm.nih.gov/37919434/>

Overall, this is a timely and interesting article, but it seems preliminary and I think so much more could be done in this interesting topic that I would encourage the authors to expand their analysis to make it even more interesting and impactful. I don't want to sound too harsh, because I really like this analysis, but I also feel more results would make the work significantly more insightful.

It is possible that some of my comments reflect misunderstandings of mine. If so then I would suggest that the authors use my misunderstandings as an indication that such points might be made clearer in the manuscript.

It my usual policy to reveal my identity to the authors:
Joao Pedro de Magalhaes.

Response to reviewers (first round):

We would like to thank the reviewers for their excellent suggestions which we believe have greatly improved our manuscript. Here we provide a brief summary of major changes in the course of this revision:

- 1- **Clarification of rationale:** We have provided a more explicit rationale for focusing on p21- and p16-positive cells, explaining why these senescence markers were prioritized over other cyclin-dependent kinase inhibitors.
- 2- **Validation using independent datasets:** We have extensively validated our findings using ten additional independent single-cell RNA-sequencing datasets. Specifically, we incorporated data from the *Tabula Muris Senis* and the *Calico murine aging cell atlas*, as suggested by the reviewers.
- 3- **Exploration of p21 and p16 expression kinetics:** In response to reviewer 2's suggestion, we investigated whether the temporal dynamics of p21 and p16 expression could explain their distinctness.
- 4- **Expansion to additional senescence datasets:** Initially, our analysis focused on the SenMayo dataset, but based on reviewer feedback, we have expanded our work to include other published senescence datasets, such as SenSig and CellAge.
- 5- **Different stimuli induce different senotypes:** To investigate whether the differences between p21+ and p16+ cell populations during aging are specific to age-related senescence or apply to other contexts, we compared our results to a mouse model of metabolic dysfunction-associated steatohepatitis (MASH), a condition characterized by increased senescence markers.
- 6- **Clarification on CyTOF data limitations:** We have also addressed the concerns raised by both reviewer 1 and reviewer 2 regarding the limitations of our single-cell proteomics CyTOF data analysis. After careful examination of the literature and consultations with leading experts in proteomics, it became clear that the availability of single-cell proteomics datasets is limited for fully exploring SASP diversity in p21+ and p16+ cells. We have now provided a more comprehensive discussion of these limitations in the revised manuscript.

We would also like to highlight that several other research groups have made comparable observations regarding the lack of overlap between p21 and p16 in their datasets. This alignment with existing literature not only reinforces our findings but also underscores the novelty and relevance of our data in the broader context of senescence research.

Detailed Reviewers' Comments:

Reviewer #1:

Remarks to the Author:

In this manuscript, Passos and colleagues have performed in-depth, single-cell transcriptomic analysis of different organs to identify senescent cells. They have focused on those cells that selectively express p21 mRNA, p16 mRNA, or both p21 and p16 mRNAs. After finding that senescent cells largely express one or the other cdk inhibitor mRNA, they identify different mRNAs encoding SASP factors present in cells that also express p21 mRNA, p16 mRNA or both mRNAs.

The authors are correct in saying that the field needs more detailed knowledge of the unique features of senescent cells. There is already widespread appreciation in the field that senescent cells are exceedingly heterogeneous, depending on cell type, stressor type and intensity, time after senescence induction, metabolic state, etc. In fact, the authors themselves have led efforts to understand this heterogeneity. Unfortunately, I am afraid the current study does not advance the field substantially, with some of the key problems identified below. Perhaps most importantly, at the present time, a manuscript that totally relies on bioinformatic analysis without validation/follow-up biology is more appropriate for a bioinformatic journal.

We sincerely appreciate the reviewer's recognition of our efforts to differentiate between p16- and p21-positive senescence across tissues. However, we would like to emphasize that our submission was categorized as an "Analysis"—a [other journal] format specifically defined as a new analysis of existing data. We are concerned that our manuscript may have been misinterpreted as an "Original article," leading to expectations of new data generation, which is outside the scope of this submission category. Our study was designed under the premise that generating new data is not required or permitted for this type of analysis, and instead, it focuses on performing new comparative analyses using pre-existing datasets, in full alignment with the "Analysis" submission format.

That said, to strengthen the robustness of our conclusions and address any concerns regarding data validation, we have included multiple additional published datasets in the revised version. We ensured that each organ analyzed now has at least two independent sc-RNAseq datasets to support our findings. In addition to brain, muscle, bone, liver, skin, and lung, we have expanded our analysis to spleen and kidney as well. This comprehensive validation includes datasets from the Tabula Muris Senis¹, the Calico murine aging cell atlas², and several other published datasets (e.g., GSE129788, GSE218300).

To our knowledge, this revised manuscript now presents the most comprehensive analysis of p21+ and p16+ -associated changes across tissues published to date. We strongly believe that our findings provide significant new insights and advance the field in a meaningful way, fitting squarely within the guidelines of the "Analysis" format.

We respectfully disagree with the reviewer's assessment that this study does not significantly advance the field. Most research on cellular senescence during aging has focused on limited markers in bulk tissues, assuming uniform p21 and p16 expression across senescent cells. This has driven efforts to identify "common" SASP factors, often overlooking the heterogeneity across cell types and tissues. To our knowledge, no study has systematically explored the variability of the SASP in p21+ and p16+ cells across multiple tissues.

Our findings reveal a striking heterogeneity in SASP profiles and a lack of overlap between p21+ and p16+ cells, challenging long-held assumptions in the field. This discovery opens new avenues for understanding senescence at a cell-type-specific level, with important implications for targeted therapies. As researchers with extensive experience in studying cell senescence over many years, we were personally surprised by the unexpected degree of heterogeneity observed in terms of the SASP, as well as the absence of overlap in the expression of p21 and p16.

Major comments

There are already numerous single-cell transcriptomic analyses of senescent cells from organs that highlight the immense heterogeneity the authors underscore here. In fact, some of those very studies are the starting point for parts of the current manuscript. This reliance on published datasets is also partly a concern: the authors do not generate much new data, and instead reanalyze earlier datasets. I appreciate the value in reanalyzing published datasets, but it does reduce the novelty of this study.

While the reviewer is correct in noting the existence of various published single-cell transcriptomic analyses of aging organs, upon careful review, we found that many of these studies did not assess senescent markers comprehensively. The few that focused on senescence merely evaluated the frequencies of p21- and p16-positive cells and lists of common SASP factors without delving into the cell-to-cell and tissue-to-tissue heterogeneity. We do not claim to be the first to observe heterogeneity in cellular senescence. However, to our knowledge, no other study has thoroughly evaluated the heterogeneity of SASP expression in p21- and p16-positive cells across multiple tissues. We would have welcomed the opportunity to conduct additional validation; however, as previously explained, this manuscript was submitted as an "Analysis" paper, aimed at extracting new insights from pre-existing datasets.

I was (briefly) excited to see that the authors attempted a limited parallel analysis of the transcriptomes by studying some proteins (CyTOF). Unfortunately, this analysis was too limited and preliminary to really contribute a valuable proteomic dimension to the study.

We agree with the reviewer that there are limitations, and we fully acknowledge them in the manuscript. We had hoped to conduct more extensive analysis of senescence and SASP markers at the protein level. However, single-cell proteomics studies focused on aging are relatively scarce. Additionally, CYTOF, like many antibody-based proteomics methodologies, is constrained by a maximum limit of around 50 antibodies and most SASP factors included are pre-selected. Consequently, our ability to explore the heterogeneity of the SASP using these approaches is restricted. Nevertheless, the CYTOF data yielded a significant finding that aligns with observations from scRNAseq datasets: namely, that at the protein level, p21 expression is notably abundant compared to p16, and both are rarely co-expressed. After consulting with proteomics experts, we were unable to identify any published datasets that allowed for direct comparison of p21+ and p16+ cells and their associated SASP. We have discussed this limitation thoroughly in the manuscript.

My enthusiasm is probably most curbed by the fact that all the entirety of the analysis is bioinformatic. The authors have not included biological validation of their computational results/predictions. If this validation had been done by *in situ* analysis of the mRNAs that are predicted to be coexpressed in a given cell (mRNAs encoding p21, p16, p21+p16 and various SASP factors) or better yet, the proteins themselves, the work would have been truly enriched.

We apologize once more for any lack of clarity. It's important to emphasize that our manuscript was submitted under the category of "Analysis," rather than "Original article." However, since we believe that a confirmation for the editors and reviewers only is helpful at this point, we include an example of spatial proteomics in the skin (Rev. Fig. 1 A-C) and transcriptomics (Xenium) from aged murine skeletal muscle (combined with H&E, Rev. Fig. 1 D) and human liver (spatial proteomics CODEX, Rev. Fig. 1E), further illustrating that there is no overlap between p16+ and p21+ cells at both transcriptomics and protein level.

Figure Rev.1- Iterative Indirect Immunofluorescence Imaging (4i) in sun-protected aged human skin (A-C), followed by Xenium analyses on H&E (D) and CODEX analyses from the and liver (E).

A) Representative image of (above) different skin cell-type markers, (below) p21 and p16; B) UMAP showing distribution of skin cell-types and C) p21, p16 and double-positive cells (dPo). Note that p21 (blue) is mostly expressed in outer and mid keratinocytes, which p16 (red) is expressed in melanocytes and fibroblasts. D) In the xenium H&E muscle analyses, p21+ cells are more abundant than p16+ cells, with no overlap. E) In the spatial proteomics (CODEX) in aged human liver, we observed some p16+ cells and a little more p21+ cells, but again no overlap between them.

Single-cell validation using flow or other single-cell methods would also have gone a long way towards strengthening the authors' hypotheses with relevant biology.

We appreciate the reviewer's suggestion. However, as previously mentioned, we encountered difficulties in identifying other published single-cell proteomics studies that would enable a thorough characterization of the SASP.

Finally, two minor but important points could be added in regard to the assembly of the manuscript:

- One is that the writing is quite 'loose', some of it reflecting a worrisome trend among data scientists to use 'gene' to refer to 'mRNA' and sometimes also to 'protein'. Here, the authors have limited their analysis to 'p21 mRNA' and 'p16 mRNA'—yet they consistently write instead

'p21' and 'p16' (the proteins). It is not clear if they are simply using incorrect nomenclature, or they are assuming that a cell that expresses p21 mRNA also expresses p21 protein. Of course, both points need attention and correction.

We are in complete agreement with the reviewer. For clarity and simplicity, we added “[...] p21 encoded by p21CIP1, for clarity and simplicity we solely use p21 and refer to p21 mRNA, but use P21 for the protein” and “... (p16 encoded by p16INK4a, for clarity and simplicity we solely use p16 and refer to p16 mRNA, but use P16 for the protein)³”.

- The other is that the graphs look pretty cool, but they are not at all intuitive or easily interpretable. I would urge the authors to use more mainstream representation that readers can follow without massive difficulty. This concern also reflects my earlier recommendation that this manuscript be published in a journal with a more bioinformatic audience.

We appreciate your perspective. Our aim was to create graphs that are accessible and easy to interpret for a broader audience, including those without a bioinformatics background. We focused on simplifying the visual presentation and ensured that each graph is clearly described in the corresponding text and figure legends for clarity and context. We respectfully disagree with the suggestion to publish this work in a bioinformatics journal. This study represents an important contribution to the field of senescence research, offering novel insights and valuable resources to the senescence community. Limiting publication to a bioinformatics-focused journal would risk missing the intended audience.

Reviewer #2:

Remarks to the Author:

The manuscript written by Dominik Saul et al. investigated p16- and p21-expressing cells in the single-cell RNA-seq datasets of multiple organs from mice and humans. The authors identified distinct secretomes associated with p16 and p21, while identifying factors common to both datasets ("core" SASP and "common" SASP). They also performed CellChat and SCENIC analyses to understand the functional significance of the transcriptomic differences. This study contains a substantial amount of data, some of which provide novel insights, but there are several fundamental questions that the authors need to address:

We thank the reviewer for recognizing that our analysis of existing data has yielded novel insights.

<Revision points>

a) Major

1) *Definition of senescent cells: The authors identified Ki-67-negative S-negative p16- and/or p21-positive cells as senescent cells in the scRNA-seq datasets. However, as the authors state in the introduction, p16 and p21 expression is not restricted to senescent cells. The authors have not confirmed that these cells are in a state of cell cycle arrest, or that these cells increase with age.*

By applying Cd45-,Ki67-,S- selection and afterwards selecting p16 and p21 positive cells, respectively, we aimed to analyze cells that were not proliferating. In the revised version, we also show age-related changes in these populations of cells, as well as determine % of cells which are in G0. We now added a plot for each tissue in which we represent the percentage of G0/G1 cells in the original dataset → Cd45- cells → Cd45-Ki67- cells → Cd45-Ki67-S- cells and p16+ as well as p21+ cells. We added a specific example in the first figure, "The selection process is depicted with an increasing percentage of G0/G1 cells across each individual step (Fig. 1b)". We added the inlay for every tissue throughout that manuscript.

In addition, the percentage of Cd45-Ki67-S- p16+ cells increases with age, as shown in brain 1 and brain 2 (Extended data Fig.1a and Extended data Fig. 2g), while the percentage of Cd45-Ki67-S- p21+ cells is restricted to specific cell types (Extended data Fig. 1b and Extended data Fig. 2h). We also added "while all p16+ cells increased with age, the increase of p21+ cells was restricted to specific clusters (Extended data Fig. 1a-b)."

Selection of cells by cell cycle stage and Ki-67 expression may improve identification, but is not sufficient to claim that a significant proportion of these cells are senescent. In particular, p21 is expressed in various non-senescent cells/states, which somewhat explains the authors' observation of p21 expression in many cells. Therefore, the authors should be more careful when using the terms "senescent" or "SASP" in the manuscript, including the manuscript title.

For example, I suggest replacing "senescent cells" with "p16/p21-expressing cells" or "p16/p21-expressing non-proliferating cells", and "SASP" should be replaced with "p16/p21-associated secretory phenotype" or "p16/p21-associated secretome", when referring to the observation in the scRNA-seq datasets.

We fully concur with the reviewer's assessment. To maintain clarity and simplicity, we intentionally avoided the term "senescence" throughout the manuscript, instead referring to p16+ and p21+ cells. At the start of our analysis, we clarify that these are specifically non-proliferating cells. Additionally, we opted to use "associated secretome" instead of SASP in the results, while for the hypotheses, we use SASP, as suggested by the reviewer. We hope this addresses the reviewer's concerns effectively.

2) Novelty: *The main findings of this study are that p16- and p21-associated secretomes are distinct in vivo, and that the secretomes are highly diverse across cell types and organs. However, as the authors note in the manuscript, it is already known that p16 and p21 are associated with distinct secretomes and biologicals in vitro. This study may be the first to characterize this distinction in vivo, but I am afraid the distinction is something that everyone would expect.*

The reviewer is correct that previous studies have demonstrated heterogeneity in cellular senescence, particularly *in vitro*. However, our findings reveal a novel and unexpected level of heterogeneity *in vivo*. Previous *in vitro* models have shown that p21 and p16 are activated with different kinetics in response to various stimuli, and both pathways are typically upregulated in replicative and stress-induced senescence. Nonetheless, the combined influence of these pathways on the secretory phenotypes remains unclear.

Many studies analyzing senescence in aged tissues have relied on qPCR-based detection of p21, p16, and common SASP factors in bulk tissues without considering specific cell types. Therefore, our analysis provides a valuable resource with significant conceptual implications for the aging research community. Recently, the development of transgenic mouse models capable of clearing p21 positive cells during aging has further underscored the distinct biological roles of p21 and p16. Studies have shown that clearing p16+ cells extends median lifespan⁴, while recent findings indicate that clearing p21+ cells extends both median and maximum lifespan⁵. Moreover, selective clearance of p21+ but not p16+ cells prevents radiation-induced osteoporosis⁶.

As the reviewer alluded to, while we cited studies highlighting functional differences between p21 and p16, these studies have primarily focused on a limited analysis of SASP components in a few tissues or using cultured cells. Our work expands upon this by providing a comprehensive examination of p21- vs p16-dependent SASP heterogeneity across multiple tissues. It also supports the hypothesis that p21 and p16 are distinct cell populations that do not share a common developmental pathway. We would submit that these concepts are, indeed, novel.

The biological/functional heterogeneity of senescent cells in different organs/cell types has also been already reported in several papers. Some of the factors discussed by the authors may not have been previously reported, but the manuscript focuses solely on describing each factor, rather than interpreting the results from a more general perspective. Therefore, more in-depth analyses and previously unknown findings are needed to claim novelty.

For example, is the difference between p16- and p21-associated secretomes parallel in vivo and in vitro (Sturmlechner I et al. 2021. Science)?

In our revised version, we broadened our analysis to include additional datasets and provide examples where we explore alternative lists of SASP factors, such as SenSig and Cell Age (also requested by reviewer 3).

While we did include an analysis of data from⁷ in our paper (See Extended Data Figure 8), it is important to recognize that this study focused specifically on mouse embryonic fibroblasts *in vitro* following overexpression of p21 and p16. Therefore, the findings may not fully reflect the SASP patterns of p21 or p16 during physiological aging *in vivo*.

Do trajectory analyses show results consistent with the "p21 in the early phase and p16 in the late phase" model (Stein GH et al. 1999. Molecular and Cellular Biology), and how do the secretory factors change at the transition?

The reviewer's point is indeed crucial, and to address this, we performed Monocle for single-cell trajectory analysis, along with scVelo or RNA Velocity for RNA velocity analysis, across all organs. Our unbiased analyses revealed that p21+ and p16+ cells follow independent developmental pathways, indicating that neither cell type originates from the other. This finding further supports their distinct identities. To our knowledge, this is the first time such an analysis has been conducted in p21+ and p16+ cells, offering novel insights.

How are the results of the CellChat/SCENIC analyses consistent (or not) with previous reports?

We appreciate the reviewer's valuable feedback. To our knowledge, no prior studies have conducted CellChat/SCENIC analyses comparing p21 and p16-positive cells across multiple tissues, making direct comparisons unavailable. Our rationale for this approach was to emphasize the unique cell communication networks and transcription factor regulation within these two cell populations.

In our CellChat analysis, the observed pathways align well with established roles in each tissue. For instance, the identification of the MAG (myelin-associated glycoprotein) pathway in p16+ cells interacting with oligodendrocytes is consistent with MAG's role in neuron-glia communication, myelination, and axon support. We have added this example to the results section. Similarly, the use of CCL and JAM pathways by p21+ cells to engage with microglia matches existing evidence of CCL chemokines in microglial activation and the role of JAM receptors in cell-cell communication and immune responses in the CNS.

While we could provide similar detailed explanations for each tissue, we aimed to avoid overcomplicating the manuscript. Our findings underscore the distinctive nature of these cell populations, warranting further experimental exploration.

How are p16/p21-related phenotypes affected by interactions with other cells and environmental cues?

We thank the reviewer for this excellent point. Our Cell-Chat analysis (Figure 5) already highlights the significant diversity in communication patterns between p21+ and p16+ cells, as well as across different cell types and tissues. Given the potential influence of environmental factors, it is likely that senescence induced by stimuli other than aging would also lead to diverse p16 and p21-related phenotypes.

To explore this further, we have now compared the development of Metabolic Dysfunction-Associated Steatohepatitis (MASH,⁸ GSE218300) with that of an aging liver¹, focusing on transcriptomic changes in p21+ and p16+ cells. Similar to the aging liver, MASH displayed minimal overlap between p21+ and p16+ cells, but the secretome profiles associated with p21 and p16 in MASH were distinct from those seen in the aging liver, highlighting the context-specific nature of senescence (Extended Data Fig.7).

3) Confirmation of the findings: I understand that the strength of this study is the observation of the heterogeneity of the potentially senescent cells in vivo. However, almost all of the results in this paper are based on single-cell RNA-seq data. As the authors may be aware, single-cell RNA-seq data are prone to multiple biases, such as cross-contamination between cell types and dropouts. Combining multiple datasets is not sufficient to guarantee that these biases are absent. Also, the difference in results between different organs may simply reflect the difference between experimental procedures, as the data from different organs came from different experiments.

We fully agree with the reviewer's assessment and acknowledge the inherent limitations of single-cell RNA sequencing (scRNA-seq) datasets. We have discussed these limitations in the discussion section of our manuscript.

We have validated our results for each organ by confirming our results using an independent scRNA-seq dataset. The CyTOF data in bone provides further support for our observation that p21 and p16 exhibit relatively low co-expression at the protein level, consistent with what we observed at the mRNA level. This seems to occur as well in other tissues. For example, unpublished data from our lab using spatial proteomics techniques (4i and CODEX) show that in aged human skin p21 and p16 expression show very little overlap and are expressed in different cell-types (See Figure Rev.1- for editor and reviewers only). However, given that 4i and CODEX are antibody based and allow for a maximum of 40 antibodies, we are not able to confirm the variability in SASP factors at the protein level.

Furthermore, if the authors are to use terms such as "core SASP", or say that the p16- and p21-associated secretory phenotypes are distinct in vivo, it is important to confirm that the secreted factors are differentially secreted, or at least show that they are differentially expressed at the protein level in vivo. Immunostaining for p16, p21, and the major putative SASP factors in multiple organs, and checking whether the results parallel with the scRNA-seq findings should be adequate to address these issues. If the authors are unable to provide such data based on other experimental methods, it should be better to use terms such as "core p16/p21-associated secretory genes" or "core putative p16/p21-associated secretome" instead of "core SASP", and to use less assertive language regarding the difference in secretomes.

We appreciate the reviewer's suggestion and have adjusted our terminology to refer to core p16/p21-associated secretory genes. We fully agree that experimentally confirming the core p16/p21-associated secretome at the protein level would significantly strengthen our findings. However, due to constraints of the "Analysis format" (see response to reviewer 1), we plan to address this validation in a subsequent research project.

Taking into account the factors identified from the SenMayo dataset, which are most commonly associated with both p21 and p16-expressing cells, we found ample evidence in the literature supporting their increased expression or secretion by senescent cells and during aging. For instance, ICAM1 is upregulated in senescent cells as reported in various contexts⁹. Similarly, IGFBP4 and IGFBP6 show increased levels in aging tissues and senescent cells¹⁰, while EDN1 (Endothelin-1) expression rises in senescent vascular cells¹¹. Additionally, CXCL16 is upregulated in aging and inflammatory conditions linked to senescence, and PLAUR (uPAR) is associated with extracellular remodeling and increased secretion by senescent cells^{12,13}. We now include these in the discussion. While our analysis aimed to be comprehensive, considering multiple tissues and datasets, we acknowledge that the identification of common factors could be influenced by the specific tissues we chose. Nonetheless, our primary objective was to illustrate the tissue-specific heterogeneity of these core p16/p21-associated secretory factors rather than defining an exhaustive list.

b) Minor

1) CyTOF: In the CyTOF analysis, three out of five factors are associated with both p16 and p21 (Fig. 2I). This seems to contradict the authors' claim that the p16- and p21-associated secretomes are highly distinct. In fact, most of the secretome factors are not shared between p16 and p21 in the bone scRNA-seq datasets (Fig. 2f, Extended Fig. 2b). As the only protein-level analysis in the study shows a result contrary to the story of the paper, the authors need to address what to make of this result. The small number of targets in CyTOF does not explain this discrepancy (the p16-p21 overlap percentage should not change if the target gene set is simply narrow down without bias).

We respectfully disagree with the assertion that the CYTOF results contradict our broader findings. Firstly, the results confirm the limited overlap between p21 and p16 expression at the

protein level, which aligns with our conclusions (we have seen the same in other proteomics datasets including in the aged skin and liver, as shown in Fig. Rev.1). For the SASP analysis, our CyTOF approach included a panel of 7 antibodies targeting SASP factors, of which only 5 showed increased expression in both p21 and p16-positive cells, and only 3 were common between them. Besides, the SASP factors were selected based on previous bulk analysis in aged bone (so that may be biased towards the common/or most abundant secreted factors).

While these results do not prove that these populations exhibit distinct secretomes at the protein level, they also do not disprove this hypothesis. This uncertainty is due to the inherent limitations of current antibody-based proteomics techniques, which rely on a restricted set of pre-selected antibodies. Unfortunately, expanding this analysis is beyond the current technological limitations.

After consulting with proteomics experts, we determined that, to the best of our knowledge, no other datasets currently provide unbiased single-cell proteomics data in aged tissues suitable for our analysis. Therefore, while we acknowledge the limitations of our current approach, these constraints are shared across the field, and we are confident in the rigor of our conclusions based on the available methodologies.

Is the difference between p16 and p21 not so pronounced at the protein level? Are the antibody targets in the CyTOF dataset biased? Or are the factors shared by p16- and p21-associated secretomes more easily captured on CyTOF? If either of these seems to be the case, what is causing this bias?

See response above. We should add that we have performed analysis in other tissues such as aged skin and liver (see data for reviewers and editor) where we observed limited overlap between p21 and p16 positive cells. In addition, we are aware of other unpublished spatial proteomics datasets in different tissues which show similar patterns.

2) Further details on cell selection: The authors have shown the proportions of cell types in the course of selecting Ki-67-negative, S-negative, p16- and/or p21-positive cells (Fig. 1b). If they could show similar figures for all the datasets in the Extended Figures, the readers would be able to better understand how the cells were narrowed down.

It would also be important to write what percentage of the cells in the parent group were selected in each selection step.

As requested, we added river diagrams for all organs in the revised manuscript.

In addition, related to the main point (1), looking at the Ki67+% and predicted cell cycle stages in CD45- p16/p21+ cells might be helpful to speculate what proportion of senescent cells are contained in the p16/p21-expressing cells.

As requested, we have provided this information in the revised version.

Reviewer #3:

Remarks to the Author:

In this work, the authors employ public scRNA-seq datasets to compare the secretomes of p16- and p21-positive senescent cells. This is a timely and important topic, the paper is well-written, and the results are interesting, yet I also feel the analysis is somewhat preliminary and more could be done, as detailed below.

We would like to thank the reviewer for their thoughtful feedback and valuable suggestions for improvement.

My major concern is that the authors call theirs a "comprehensive analysis", including in the abstract, yet it doesn't feel that way because several major datasets are not included and the work leaves several questions open. In particular, the authors don't include the larger scRNA-seq datasets of aging I'm aware of, namely the Tabula Muris Senis and the Calico murine aging cell atlas:

We thank the reviewer for the suggestion. In the revised version of our manuscript, we have incorporated ten additional datasets. By integrating these independent datasets, confirmed the generalizability of our findings and evaluated the reproducibility of our results across different experimental contexts.

Related to the above, I couldn't find information on why certain data sets were selected, but not others. I think either the authors need to perform a comprehensive analysis incorporating at least the major publicly available scRNA-seq datasets or explain their criteria for data selection.

To minimize potential variability in our single-cell RNA sequencing (scRNA-seq) datasets—such as differences in mouse age, animal facilities, cell isolation methods, and data processing techniques—our first approach was to focus on studies conducted and published by our co-authors. These studies, which include murine brain¹⁴ and skeletal muscle¹⁵, were carried out under controlled conditions, ensuring consistency in mouse strains, housing, and experimental protocols. Additional independent methods (q-PCR, RNA-ISH and IF) were also employed to characterize and validate the senescent cell burden with age.

However, in response to the reviewer's feedback and to enhance the generalizability of our findings, we have now incorporated several additional datasets, including the Tabula Muris Senis dataset¹, Calico Murine Aging Cell Atlas², another brain dataset (GSE129788 from¹⁶), and a liver dataset (GSE218300 from⁸). We also replicated all analyses across different datasets from the same tissue, consistently observing similar trends and findings. This broader approach provides additional robustness and generalizability to our conclusions.

The main results are that there are substantial differences between the secretomes of p16- and p21-positive senescent cells. I think this is an interesting observation, but it also leaves a few open questions. In particular, I couldn't find results on the number or frequency of p16- and p21-positive senescent cells, and how these change with age.

As requested, we added the percentage of Cd45-,Ki67-,S- p16+ / p21+ cells in Fig. 1 and for each other organ. For the brain, we added the percentages of Cd45-/Ki67-/S- and p16+ cells or p21+ cells in young vs. old in Extended Data Fig. 1 a and b and Extended data Fig. 2 g and h, respectively. Interestingly, while p16+ cells increased with age, this was only true for some clusters of p21+ cells. Since in most datasets, young and old samples were not available, and to not further flood the extended data figures (as for now, there are thirteen), we did not include these for all other organs.

The original analysis performed on the Calico data found very few cells expressing markers of senescence and no significant changes in aged tissues, so determining whether the percentage of cells exhibiting markers of senescence changes with age would be important.

We now provide the percentage of p21 and p16 positive cells with age in the brain in Extended Data Fig. 1a and b and in Extended Data Fig. 2g and h. Please see above. Since specifically requested, we calculated the percentages of young vs. old in all Calico datasets.

The reviewer is correct in noting that the senescence markers p21 and p16 did not exhibit significant age-dependent increases in the Calico dataset. However, these findings align with observations from other datasets, where p21 is consistently found to be more abundant than p16. While p16 does show a tendency to increase with age in specific organs such as the kidney and spleen, the changes are not statistically significant, likely due to the small sample sizes (n=4 for young and n=3 for old). Although p21 does not demonstrate a significant age-related increase across the overall cell population, it is possible that it may rise with age in particular p21+ cell types.

We acknowledge the inherent limitations of single-cell RNA sequencing datasets, especially in the context of studying senescence, which may contribute to the observed variability across studies. Factors such as poor cell quality or viability during sample preparation can skew results, underscoring the importance of careful experimental design and data interpretation.

The authors limit their analysis to the secretome. I wonder why not perform a broader analysis of gene expression changes between p16- and p21-positive senescent cells?

We decided to focus on the secretome, given ongoing efforts within the senescence community to identify signatures of SASP factors across tissues^{17,18}. Our initial expectation was that we would find more commonalities in SASP profiles in p21 and p16 positive cells across tissues. Surprisingly, we found that only a limited number of factors were common, which we believe has important conceptual implications. Having said that, we have now extended our analysis to include other senescence associated datasets such as SenSig and CellAge, which include only around 15% of secreted factors as opposed to SenMayo, which includes c.a. 73% (according to the Human Protein Atlas).

In addition, I think it would be good to validate at least some results in another data set of senescence signatures, like for example our CellAge database:

<https://genomics.senescence.info/cells/>

We thank the reviewer for the suggestion, as suggested we now performed this analysis using the CellAge database. We analyzed all datasets with not only SenMayo, but also SenSig and CellAge, added as Fig. 4f-h and Fig. 4i-k, respectively.

Also, why limit the analysis to p16 and p21 as markers of senescence? For instance, a recent study employed a transfer learning for signatures to identify senescent cells from single cell data:

<https://link.springer.com/article/10.1007/s11357-023-00785-7>

Our primary objective in this study was to assess and underscore the distinctiveness of p16 and p21 during aging *in vivo*, given their longstanding use as indicators of senescence. We specifically focused on p21 and p16 because they are critical regulators of the cell-cycle arrest that defines senescence. Furthermore, recent guideline papers in *Cell* and *Nature Rev Mol Cell*

Biol authored by leading experts in the senescence research community have strongly recommended including these two CDK inhibitors (CDKis) in any *in vivo* analysis of senescence^{19,20}. While it is possible that other cyclin-dependent kinase inhibitors (CDKis) may also contribute to senescence (we acknowledge that in the manuscript), incorporating them into our analysis would significantly increase the complexity of our manuscript and decrease its clarity.

In the revised version, we will also use the SenSig signature demonstrated in the aforementioned manuscript and investigate how it is associated with p21 and p16.

See above.

A couple of minor comments:

Line 112: "when we analyzed an extensive list of secreted proteins not limited to the known SASP14 plus SenMayo" - what does this mean exactly? How many secreted proteins were analyzed?

The secreted 1891 proteins were taken from the human protein atlas (https://www.proteinatlas.org/search/protein_class:Predicted+secreted+proteins+). We will make sure this is clearer in the manuscript.

Line 338: why not make the code available on GitHub?

We are pleased to accommodate this request with the whole code for each figure after acceptance.

Lastly, I think the work could be better placed in the context of previous findings. As above-mentioned, for instance, the Calico analysis studied senescent cells from single cell data. Our analysis of intercellular communication in aging using scRNA-seq data also revealed known SASP factors associated with the increase in inflammation with age:

<https://pubmed.ncbi.nlm.nih.gov/37919434/>

We carefully reviewed this work and discuss it in relation to our findings: "The cell-cell-communication patterns we found have been similarly detected by Lagger et al. combining the Tabula Muris Senis with the Calico murine aging cell atlas²¹. While our CellChat analyses detected a reduction of the collagen signaling in four tissues, this downregulation was similarly found by Lagger et al. Likewise, the reduction of App that was almost consistently found in our analyses has been highlighted in male tissues and connected to a variety of tissue-specific diseases in the work by Lagger et al."

Overall, this is a timely and interesting article, but it seems preliminary and I think so much more could be done in this interesting topic that I would encourage the authors to expand their analysis to make it even more interesting and impactful. I don't want to sound too harsh, because I really like this analysis, but I also feel more results would make the work significantly more insightful.

It is possible that some of my comments reflect misunderstandings of mine. If so then I would suggest that the authors use my misunderstandings as an indication that such points might be made clearer in the manuscript.

We sincerely appreciate your candid evaluation of our manuscript and are grateful for the insightful suggestions, which we believe have significantly strengthened our work. We hope that the inclusion of additional datasets, along with our clarifications, adequately addresses your

concerns and enhances the overall quality of the paper. Thank you once again for your invaluable input.

Sincerely,

Dominik Saul
Sundeep Khosla
Joao Passos

References:

- 1 A single-cell transcriptomic atlas characterizes ageing tissues in the mouse. *Nature* **583**, 590-595 (2020). <https://doi.org/10.1038/s41586-020-2496-1>
- 2 Kimmel, J. C. *et al.* Murine single-cell RNA-seq reveals cell-identity- and tissue-specific trajectories of aging. *Genome Res* **29**, 2088-2103 (2019). <https://doi.org/10.1101/gr.253880.119>
- 3 Gorgoulis, V. *et al.* Cellular Senescence: Defining a Path Forward. *Cell* **179**, 813-827 (2019).
- 4 Baker, D. J. *et al.* Naturally occurring p16Ink4a-positive cells shorten healthy lifespan. *Nature* **530**, 184-189 (2016). <https://doi.org/10.1038/nature16932>
- 5 Wang, B. *et al.* Intermittent clearance of p21-highly-expressing cells extends lifespan and confers sustained benefits to health and physical function. *Cell Metab* **36**, 1795-1805.e1796 (2024). <https://doi.org/10.1016/j.cmet.2024.07.006>
- 6 Chandra, A. *et al.* Targeted clearance of p21- but not p16-positive senescent cells prevents radiation-induced osteoporosis and increased marrow adiposity. *Aging Cell* **21**, e13602 (2022). <https://doi.org/10.1111/accel.13602>
- 7 Sturmlechner, I. *et al.* p21 produces a bioactive secretome that places stressed cells under immunosurveillance. *Science* **374**, eabb3420 (2021). <https://doi.org/10.1126/science.abb3420>
- 8 Bendixen, S. M. *et al.* Single cell-resolved study of advanced murine MASH reveals a homeostatic pericyte signaling module. *J Hepatol* **80**, 467-481 (2024). <https://doi.org/10.1016/j.jhep.2023.11.001>
- 9 Gorgoulis, V. G. *et al.* p53-dependent ICAM-1 overexpression in senescent human cells identified in atherosclerotic lesions. *Lab Invest* **85**, 502-511 (2005). <https://doi.org/10.1038/labinvest.3700241>
- 10 Moiseeva, V. *et al.* Senescence atlas reveals an aged-like inflamed niche that blunts muscle regeneration. *Nature* **613**, 169-178 (2023). <https://doi.org/10.1038/s41586-022-05535-x>
- 11 Alcalde-Estévez, E. *et al.* Endothelin-1 induces cellular senescence and fibrosis in cultured myoblasts. A potential mechanism of aging-related sarcopenia. *Aging (Albany NY)* **12**, 11200-11223 (2020). <https://doi.org/10.18632/aging.103450>
- 12 Amor, C. *et al.* Senolytic CAR T cells reverse senescence-associated pathologies. *Nature* **583**, 127-132 (2020). <https://doi.org/10.1038/s41586-020-2403-9>
- 13 Coppé, J. P. *et al.* Senescence-associated secretory phenotypes reveal cell-nonautonomous functions of oncogenic RAS and the p53 tumor suppressor. *PLoS Biol* **6**, 2853-2868 (2008). <https://doi.org/10.1371/journal.pbio.0060301>
- 14 Ogrodnik, M. *et al.* Whole-body senescent cell clearance alleviates age-related brain inflammation and cognitive impairment in mice. *Aging Cell* **20**, e13296 (2021). <https://doi.org/10.1111/accel.13296>
- 15 Zhang, X. *et al.* Characterization of cellular senescence in aging skeletal muscle. *Nat Aging* **2**, 601-615 (2022). <https://doi.org/10.1038/s43587-022-00250-8>

- 16 Ximerakis, M. *et al.* Single-cell transcriptomic profiling of the aging mouse brain. *Nat Neurosci* **22**, 1696-1708 (2019). <https://doi.org/10.1038/s41593-019-0491-3>
- 17 Schafer, M. J. *et al.* The senescence-associated secretome as an indicator of age and medical risk. *JCI Insight* **5** (2020). <https://doi.org/10.1172/jci.insight.133668>
- 18 Oguma, Y. *et al.* Meta-analysis of senescent cell secretomes to identify common and specific features of the different senescent phenotypes: a tool for developing new senotherapeutics. *Cell Commun Signal* **21**, 262 (2023). <https://doi.org/10.1186/s12964-023-01280-4>
- 19 Ogrodnik, M. *et al.* Guidelines for minimal information on cellular senescence experimentation in vivo. *Cell* **187**, 4150-4175 (2024). <https://doi.org/10.1016/j.cell.2024.05.059>
- 20 Suryadevara, V. *et al.* SenNet recommendations for detecting senescent cells in different tissues. *Nat Rev Mol Cell Biol* (2024). <https://doi.org/10.1038/s41580-024-00738-8>
- 21 Lagger, C. *et al.* scDiffCom: a tool for differential analysis of cell-cell interactions provides a mouse atlas of aging changes in intercellular communication. *Nat Aging* **3**, 1446-1461 (2023). <https://doi.org/10.1038/s43587-023-00514-x>

Reviewers' Comments after resubmission of revised version:

Reviewer #1 (Remarks to the Author): defer to editor

Reviewer #2 (Remarks to the Author):

I appreciate the authors' effort in adding a substantial amount of data.

However, the fundamental problem has not been solved.

No confirmation experiments have been conducted (due to the "analytical format" problem), and terms are used carelessly. Even if it is positioned as a preliminary and exploratory analysis, the new findings in this paper are considered insufficient for publication in [other journal].

1) Definition of senescent cells

The authors added a figure to show the age-related increase in Cd45-Ki67-S- p16+ cells (although the difference is only significant in the brain datasets).

However, there is no additional experimental data to show that the cells are in a state of cell cycle arrest (probably due to the "Analysis format" issue mentioned by the authors). As the authors may be aware, it is difficult to determine whether the cells are senescent or not, based solely on omics data.

I appreciate the authors' carefulness in avoiding the use of the word "senescence" throughout the manuscript. However, since there is no experimental evidence to confirm that the cells identified in the analyses contain a significant proportion of senescent cells, terms such as "SASP" and "senotypes" also should not be used in the title or in the paragraphs except for the Introduction. In the comments, the authors state, "while for the hypotheses, we use SASP", but this is also to be avoided outside of the Introduction section.

For example, the following parts in the abstract should be altered:

> we identified a limited set of 55 shared "core" SASP factors that may drive common senescence-related functions.

They cannot be called "SASP factors".

> Our study underscores the extraordinary diversity of cellular senescence and the SASP, This is an overstatement. Since we are not sure if the identified cells are senescent cells or not, the results are clearly insufficient to make this claim.

All the other parts of the manuscript should be rewritten as well.

2) Novelty

I appreciate the authors for conducting the analysis I suggested. I would like to comment on them first:

The authors analyzed the data set from Sturmlechner I et al. (Extended Data Figure. 9).

However, this is not the analysis I wanted to propose. The difference between p16- vs. p21-secretome is already clear from the original paper. What I was interested in was whether the difference between the Sturmlechner I et al. dataset and the authors' results was parallel. Were there common factors that differed between p16 vs. p21? What were the common features in factors related only to p16/p21 in each analysis?

I also appreciate the authors for performing the trajectory analyses on p16+ and p21+ cells. It is interesting that the trajectories of p16+ and p21+ cells do not align. The issue of the developmental distinction between p16+ vs. p21+ senescent cells is important and should be an important paper if the finding is supported by solid experimental evidence.

I appreciate the discussion of the CellChat/SCENIC analyses and interactions with the environment. As the authors noted, the use of MAG/CCL/JAM pathways by p16/p21-expressing non-proliferating cells may be somewhat consistent with the previous findings.

Having said the above, I am afraid that the manuscript still lacks novelty. The authors made several novelty points in the rebuttal comments, but I found none of them convincing. The authors commented as follows:

However, our findings reveal a novel and unexpected level of heterogeneity in vivo. I do not consider the observed heterogeneity to be "novel and unexpected". I believe that most experts in the field share this view. There is also no objective evidence to support this claim.

Therefore, our analysis provides a valuable resource with significant conceptual implications for the aging research community. I think that sufficient experimental verification will be required for this comment.

It also supports the hypothesis that p21 and p16 are distinct cell populations that do not share a common developmental pathway. We would submit that these concepts are, indeed, novel. The results are interesting, but too weak to say that they actually support the hypothesis. Trajectory analyses are heavily influenced by parameters and the bias in the data set, and they lack reliability without experimental evidence. The distinction of developmental trajectories is not conceptually novel either.

I mentioned in the first review comments that some of the results "provide novel insights", but this does not really count as sufficient novelty since the results have not been confirmed in additional experiments.

The manuscript as it stands is just a list of unverified hypotheses with overstatements.

3) Confirmation of the findings

I understood the limitations of the "Analysis format". Since the experiments cannot be performed and the CyTOF dataset provides limited data, there is no way to prove that the secretory factors are indeed differentially secreted or produced.

I acknowledge that some of the secretory factors observed have already been reported as SASP factors in the previous studies e.g. SenMayo. However, this of course does not guarantee that the factors appearing in the results actually constitute the SASP.

I appreciate the authors' careful use of the term "p16/p21-associated secretory genes" instead of the "SASP". Since the findings are not confirmed, terms such as "secretome", "secretory phenotype", or "secretory profile" should not be used outside the introduction section, as I stated in the first review comment.

b) Minor

I appreciate that the authors describe the limitations of the experimental setting with respect to the CyTOF data set.

As the authors stated, "While these results do not prove that these populations exhibit distinct secretomes at the protein level, they also do not disprove this hypothesis". I agree with this. Therefore, some statements in the manuscript such as: "Nevertheless, we identified proteins such as Serpine1 ... reinforcing the existence of distinct secretory profiles" are inappropriate. The CyTOF results in no way confirm the existence of distinct secretory profiles.

Reviewer #3 (Remarks to the Author):

The authors addressed my concerns and the manuscript is improved. It is not clear to me, however, whether the authors incorporated additional data sets as part of their meta-analysis or as a subsequent validation step? It would be helpful, in fact, to have a table with a list of all the data sets used in the analysis. Line 414 refers to the data processing but I could not find in the Methods a list of datasets used.

Response to reviewers second round:

Summary of Reviewer 2's comments: Reviewer 2 acknowledges the substantial effort made to expand the analysis but maintains that the manuscript lacks sufficient novelty. They express concern about the use of terms like "senescence," "SASP," and "secretory phenotype" given the absence of direct validation that the identified cells are truly senescent. While they appreciate the revised language and inclusion of trajectory and interaction analyses, they argue that the findings remain largely descriptive and insufficiently novel without experimental follow-up.

Our response:

Definition of senescent cells: We respectfully disagree with the reviewer's suggestion to avoid using the terms "SASP factors" or "senotypes" in the manuscript. The cell populations we analyzed—those expressing Cdkn1a (p21) or Cdkn2a (p16), lacking proliferation markers such as Ki67, and exhibiting elevated expression of multiple secretory and inflammatory genes—meet widely accepted operational definitions for identifying senescent-like cells in vivo. This approach is supported by recent expert consensus statements (Ogrodnik et al., Cell, 2024; Suryadevara et al., Nat Rev Mol Cell Biol, 2024), which explicitly acknowledge that definitive experimental confirmation of cellular senescence in vivo is rarely feasible. These statements emphasize that the combined presence of cell cycle arrest markers (such as p21 or p16), lack of proliferation, and expression of secretory or inflammatory mediators constitutes a reasonable and widely used strategy for inferring senescence in complex tissues.

For example, Ogrodnik et al. note: "Evidence of the expression of at least three senescence markers representing different properties of cellular senescence in a given tissue strongly supports the notion that the observed phenomena are indeed related to cellular senescence. At least one of these markers needs to indicate stable cell cycle inhibition in the form of increased p21 or p16 expression... Ideally, multiple markers should be co-detected in the same cell at the same time."

Despite this, we have taken care throughout the manuscript to avoid overinterpretation based on the suggestion from the reviewer. We refer to "p21⁺" or "p16⁺" populations and describe their transcriptional profiles as "p21- or p16-associated secretomes" rather than using the term SASP indiscriminately. Nevertheless, we do use the terms "senotype" and "SASP" in the title, introduction, and summary to communicate the broader biological context and relevance of our findings. Omitting these terms entirely would, in our view, obscure the senescence-related framework in which these molecular phenotypes operate and reduce the utility of our study as a resource for the aging research community.

We fully agree with the reviewer that defining senescence in vivo remains challenging, but we believe our approach is transparent, consistent with current best practices, and appropriately cautious in interpretation.

Novelty:

We respectfully disagree with the reviewer's assessment that our study lacks novelty. While it is well established that SASP expression is heterogeneous, to our knowledge, no prior study has systematically examined this heterogeneity across multiple tissues by directly comparing non-proliferating p21⁺ and p16⁺ cells using publicly available single-cell RNA-seq datasets. Importantly, we are careful not to claim that we are the first to report heterogeneity in senescence. We explicitly acknowledge and cite foundational work from the Demaria lab and others who demonstrated transcriptional heterogeneity in senescent cells induced in vitro across different cell types.

What sets our study apart is its scope and in vivo relevance: we perform a cross-tissue, comparative analysis of p21⁺ and p16⁺ cells in multiple murine and human organs, uncovering both conserved and tissue-specific features associated with these cell states. This provides not only a valuable resource but also a conceptual framework that we believe is timely and informative for the field.

The reviewer suggests that our findings would not be considered novel by most experts, yet provides no specific evidence or citations to support this claim. In contrast, the prevailing assumption in the literature, primarily based on in vitro studies, is that p21 is an early stress response marker, with p16 acting downstream to maintain long-term arrest, implying a sequential relationship within the same cells. Our data challenge this view by showing that p21⁺ and p16⁺ non-proliferating cells are largely non-overlapping in vivo, exhibit distinct transcriptional profiles, and follow independent developmental trajectories across tissues. As researchers deeply involved in senescence biology, we found these results both unexpected and conceptually important.

We appreciate the reviewer's feedback and are happy to further temper any language that may be perceived as overstating our conclusions. However, we stand by the novelty and relevance of our findings. We believe our work fills a critical gap by shifting attention toward the in vivo identity and diversity of senescent-like cells and offers a foundation for future mechanistic and translational investigations. We hope the reviewer will reconsider the significance of this contribution in light of these clarifications.

Specific comment: "I was interested in was whether the difference between the Sturmlechner I et al. dataset and the authors' results was parallel. Were there common factors that differed between p16 vs. p21? What were the common features in factors related only to p16/p21 in each analysis?"

There is some overlap between the transcriptomic analyses from Sturmlechner et al., which were performed in mouse embryonic fibroblasts, and our single-cell RNA-seq data. Among the p16-associated SASP factors, Mmp2 and Il18 were consistently upregulated in both datasets. Similarly, for the p21-associated SASP, Serpine2 was upregulated across both studies.

3) Confirmation of the findings: I understood the limitations of the "Analysis format". Since the experiments cannot be performed and the CyTOF dataset provides limited data, there is no way to prove that the secretory factors are indeed differentially secreted or produced. I acknowledge that some of the secretory factors observed have already been reported as SASP factors in the previous studies e.g. SenMayo. However, this of course does not guarantee that the factors appearing in the results actually constitute the SASP.

I appreciate the authors' careful use of the term "p16/p21-associated secretory genes" instead of the "SASP". Since the findings are not confirmed, terms such as "secretome", "secretory phenotype", or "secretory profile" should not be used outside the introduction section, as I stated in the first review comment.

We appreciate the reviewer's careful attention to terminology and understand the importance of precision in describing transcriptomic findings. We carefully considered various alternatives to the term "SASP" and ultimately chose "associated secretory genes" or "associated secretome" because the genes we refer to are transcripts that encode proteins commonly secreted by cells and frequently reported in prior studies as components of senescence-associated secretomes. Importantly, we would like to note that most published SASP gene sets, including those widely cited in the field, have also been derived primarily from transcriptomic data, and very few have been comprehensively validated at the protein level across tissues. Therefore, our decision to use "associated secretory genes" reflects both our cautious interpretation of the data and the current limitations in the field.

We remain open to further refining the terminology if the editor or reviewers feel it would improve clarity, but we believe our current phrasing strikes an appropriate balance between accuracy and biological relevance.

Reviewer #3 (Remarks to the Author):

The authors addressed my concerns and the manuscript is improved. It is not clear to me, however, whether the authors incorporated additional data sets as part of their meta-analysis or as a subsequent validation step? It would be helpful, in fact, to have a table with a list of all the data sets used in the analysis. Line 414 refers to the data processing but I could not find in the Methods a list of datasets used.

We thank the reviewer for their positive feedback and helpful suggestion. The additional datasets were incorporated as a subsequent validation step to test the generalizability of our findings across independent cohorts. We agree that clearly distinguishing between the primary analysis and validation datasets will improve transparency and clarity.

In response, we include a new table 1 listing all datasets used, along with details such as tissue of origin, source publication, accession numbers. We will also update the Methods section to explicitly reference this table and clarify the role of each dataset in the overall study. We appreciate the reviewer's careful reading and believe these additions will make the manuscript more accessible to readers.

Additional data available for inclusion upon request:

While it is possible that the distinction between p21⁺ and p16⁺ cell populations observed in our single-cell RNA sequencing data could be influenced by technical limitations such as cell loss during tissue dissociation, we have now taken additional steps to address this concern. Specifically, we have performed spatial transcriptomics (MERFISH) in aged human skin. Our MERFISH data from aged skin independently confirms that p21⁺ and p16⁺ cells are largely distinct populations, each exhibiting a unique associated secretome (**see Supplementary Figure 1**). If reviewers consider it necessary, we would be glad to include this dataset in the revised manuscript to further substantiate our findings.

Supplementary Figure 1- Multiplexed Error-Robust Fluorescence In Situ Hybridization (MERFISH) analysis on human skin samples. A) Cell types within the human skin sample. **B)** P16+ (red) and P21+ (blue) cells are distinct with hardly any overlap between these markers. **C)** P16+ and P21+ associated SASP within the MERFISH dataset.

We have also examined p21 and p16 expression at the protein level in human tissues using immunofluorescence. Antibody specificity was rigorously validated using p21 and p16 knockout cell lines in our laboratory (not shown). Across multiple tissue types, we consistently observe a lack of co-localization between p21 and p16 proteins. This pattern is evident not only in aged human skin, but also in lung sections from patients with idiopathic pulmonary fibrosis (IPF) and chronic obstructive pulmonary disease (COPD), both of which are associated with increased expression of senescence-associated markers. While we recognize that the presence of p21 or p16 alone does not define a senescent state, the consistent lack of overlap between these markers, observed at both the RNA and protein levels across tissues, underscores their distinct biological roles. If deemed helpful, we are happy to include representative examples from these human tissues in the revised manuscript (see **Supplementary Figure 2**).

A

Human aged skin (sun protected)

B

Human lung (COPD and IPF)

Supplementary Figure 2. Immunofluorescence analysis of p21 and p16 expression in human skin and lung.

A) Representative image of aged human skin showing distinct cellular populations expressing p16 (red) and p21 (green), with minimal spatial overlap.

B) Representative images of human lung tissue from patients with chronic obstructive pulmonary disease (COPD) and idiopathic pulmonary fibrosis (IPF), demonstrating similarly distinct localization of p16-positive (red) and p21-positive (green) cells.

These findings support the existence of separate p16⁺ and p21⁺ cell populations in tissues.

Dear Joao,

Thank you again for the submission of your amended manuscript (EMBOJ-2025-121830) to The EMBO Journal. Please accept again my apologies for the unusual protraction with the assessment process due to delayed expert input. We have carefully evaluated your manuscript and the point-by-point response provided to the referee concerns that were raised during review at a different journal. In addition, and as mentioned before, we decided to involve two arbitrating experts to evaluate the revised version of your work, with respect to technical robustness, conceptual advance and overall suitability of your work for publication in The EMBO Journal.

As you will see from the arbitrating comment enclosed below, the advisors are in favour of the work stating the interest and value of your integrative analyses and results and therefore supportive of publication at The EMBO Journal. They also offer constructive input on how to improve the manuscript for a final revision by integrating complementary validating data.

We are thus pleased to inform you that we can offer to swiftly move forward towards acceptance of this work at The EMBO Journal as a data resource.

Please consider below input carefully and revise your manuscript accordingly by the additional data from your rebuttal response or textual adjustments and introducing caveats where appropriate.

Also, we now need you to take care of a number of minor issues related to formatting and data annotation, which I will share shortly in a separate message, together with additional changes and requests by our production team and information regarding Source Data provision.

Please submit a revised version of the manuscript at your earliest convenience using the link enclosed below, addressing the

advisors' comments.

As you might remember from previous experience, every paper at the EMBO Journal now includes a 'Synopsis', displayed on the html and freely accessible to all readers. The synopsis includes a 'model' figure as well as 2-5 one-short-sentence bullet points that summarize the article. I would appreciate if you could provide this figure and the bullet points.

Thank you again for giving us the chance to consider your manuscript for The EMBO Journal, I look forward to hearing from you and receiving your final revised version of the manuscript.

Best regards,

Daniel

EMBOJ-2025-121830

Arbitrating advisor #1's comment:

I have reviewed the manuscript and, overall, I concur with the reviewers' concerns, particularly regarding the lack of experimental validation. The study reanalyses several publicly available single-cell RNA-seq datasets, including some generated by the authors.

The results suggest that p16+ and p21+ cells represent distinct populations. This is interesting and has some novelty, especially in the context of in vivo models. However, the conclusions are often stated too strongly without experimental validation. For example, the claim that there is "no evidence of direct transitions between these two states" (summary) is based solely on pseudotime analysis—a computational inference that offers limited insight without supporting experimental data.

From what I can tell, the single-cell datasets primarily use the 10x Chromium platform, which is known to miss lowly expressed genes, and p16 transcripts are often low. Therefore, the absence of p16 expression in a subset of p21+ cells warrants further experimental validation. The authors appear to have spatial profiling data, which could help address this concern.

Given that the journal accepts data analysis-focused submissions, it is especially important to avoid overstatements. Nevertheless, some validations (FISH, IF, Xenium) would be necessary.

Arbitrating advisor #2's comment:

I have now read in detail the manuscript and the two rounds of comments and responses by reviewers and authors.

I have to say that I find merit and novelty in this paper as a resource. The analyses performed, especially after the additions to address the reviewer's comments, are very thorough and compelling. It is true that the ideas of heterogeneity and separate trajectories of p16+ and p21+ senescent cells are not novel, but the depth and breadth of analysis provided here is significantly higher compared to previous studies and I anticipate that this will be a reference study for future research.

The addition of the MERFISH and IF data offered by the authors will add an orthogonal layer of validation that will further improve the paper.

In summary, I am in favor of publication of this work and I think that it is very much aligned with the expected quality for an EMBO J. paper.

Dear Joao, dear Dominik,

Further to below, please find the mentioned additional formatting requirements for this article enclosed below.

Please let us know any time should there be additional questions related.

We look forward to your final revision.

Best regards,
Daniel

EMBOJ-2025-121830

>> Please add up to five keywords to your study.

>> Author Contributions: Remove the author contributions information from the manuscript text. Note that CRediT has replaced the traditional author contributions section as of now because it offers a systematic machine-readable author contributions format that allows for more effective research assessment. and use the free text boxes beneath each contributing author's name to add specific details on the author's contribution.

More information is available in our guide to authors.
<https://www.embopress.org/page/journal/14602075/authorguide>

>> Please remove the "Emails" list from the manuscript text. Please let us know if the intention is to have every author's ORCID number linked - if this is the case we will send emails to all of them with guidance on how to do so.

>> Section order should be corrected as follows: title page with complete author information, abstract, keywords, introduction, results, discussion, methods, data availability section, acknowledgements, disclosure and competing interests statement, references, main figure legends, tables, expanded figure legends.´.

>>Rename the 'Summary' to 'Abstract'.

>> Adjust the title of the 'Competing Interests' section to 'Disclosure and Competing Interests Statement'.

>> Provide a completed Author Checklist.

>> Add complete annotation of animal husbandry -mouse ethics as well as information on human patient consent to the Methods and adjust the Author Checklist accordingly.

>> References: adjust the reference format to EMBO Journal format, 10 authors et al, and place References after the Disclosure and competing interests statement, before figure legends. please remove the DOIs.

>>Figures in separate files: Please remove the figures from the manuscript text and upload them as individual, high resolution figure files. Please place the figure legends at the end of the manuscript text. Please rename the supplementary figures "Figure EV1" etc., remove them from the manuscript text and upload them as individual, high resolution figure files. Please place the legends after the main figure legends, under the heading "Expanded View Figure Legends".

>> Data availability section: please add Data availability section and integrate the Code availability section. Deposit the code at suitable repository and make it publicly available.

>> Figure callouts: Please ensure that figures are called out in sequential order in the main text. Currently, Figure EV4A and EV5A are called out before Figure EV3B; all panels should be cited for Figure EV4 and EV5.

>> Add a Reagents and Tools table to the Methods section, as a separate file using the existing template in the Guide For Authors, listing key reagents, experimental models, software and relevant equipment.

>> Please provide a completed source data and a source data checklist for the study as requested by the separate e-mail.

>>Funding: please enter the following funding information in the list of funders in our online system: 'R01AG82708, HR-GRO-23-1199144-8, R01 AG063543, R01AG07651, U54 AG079754, R0121AG055529, U54 AG076041, U19 AG056278 and NRF grant (MSIT, NRF-2020M3A9D8038014)'.

>> Consider additional changes and comments from our production team as indicated below:

- DAS

Please note that the data availability statement is not provided in the manuscript.

- Figure legends:

1. Please define the annotated p values ****/***/**/* as well as provide the exact p-values for the same in the legend of extended data figures 12C, D, E, G as appropriate.

2. Please indicate the statistical test used for data analysis in the legends of extended data figures 1A, B; 2G, H; 12C, D, E, G

3. Please note that information related to n is missing in the legends of extended data figures 1A, B; 2G, H; 12 C-E
4. Please note that the error bars are not defined in the legends of extended data figures 1A, B; 2G, H; 12G.

EMBOJ-2025-121830

Arbitrating advisor #1's comment:

I have reviewed the manuscript and, overall, I concur with the reviewers' concerns, particularly regarding the lack of experimental validation. The study reanalyses several publicly available single-cell RNA-seq datasets, including some generated by the authors.

The results suggest that p16+ and p21+ cells represent distinct populations. This is interesting and has some novelty, especially in the context of in vivo models. However, the conclusions are often stated too strongly without experimental validation. For example, the claim that there is "no evidence of direct transitions between these two states" (summary) is based solely on pseudotime analysis—a computational inference that offers limited insight without supporting experimental data.

From what I can tell, the single-cell datasets primarily use the 10x Chromium platform, which is known to miss lowly expressed genes, and p16 transcripts are often low. Therefore, the absence of p16 expression in a subset of p21+ cells warrants further experimental validation. The authors appear to have spatial profiling data, which could help address this concern.

Given that the journal accepts data analysis-focused submissions, it is especially important to avoid overstatements. Nevertheless, some validations (FISH, IF, Xenium) would be necessary.

We are very grateful for these thoughtful comments and agree that experimental validation is important to support our computational findings. In the revised version, we have taken several steps to address these concerns. First, we incorporated an additional independent dataset, MERFISH from aged, sun-protected human skin, which preserves spatial information and independently confirms that p21- and p16-expressing cells represent distinct and non-overlapping populations. Second, we performed immunofluorescence analyses in aged sun-protected human skin as well as in diseased human lung tissue (COPD and IPF), which again revealed limited overlap between p21 and p16 expression at the protein level. These experimental validations, now included in the manuscript (see Figures 3 and Extended data Figure 10), provide orthogonal support for the computational analyses and reinforce our conclusion that p21+ and p16+ cells are distinct entities. We have also carefully revised the text throughout to avoid overstatements and to clarify that our trajectory analyses indicate computationally that

no direct transitions between these two states are detected, while experimental confirmation further strengthens the case for their distinctiveness.

Arbitrating advisor #2's comment:

I have now read in detail the manuscript and the two rounds of comments and responses by reviewers and authors.

I have to say that I find merit and novelty in this paper as a resource. The analyses performed, especially after the additions to address the reviewer's comments, are very thorough and compelling. It is true that the ideas of heterogeneity and separate trajectories of p16+ and p21+ senescent cells are not novel, but the depth and breadth of analysis provided here is significantly higher compared to previous studies and I anticipate that this will be a reference study for future research.

The addition of the MERFISH and IF data offered by the authors will add an orthogonal layer of validation that will further improve the paper.

In summary, I am in favor of publication of this work and I think that it is very much aligned with the expected quality for an EMBO J. paper.

We sincerely thank the advisor for the supportive and constructive evaluation of our work. We are pleased that the added analyses and validations strengthened the manuscript and that the study is considered a valuable resource for the field.

Dear Joao,

Thank you for submitting the revised version of your manuscript. I have now evaluated your amended manuscript and concluded that the remaining minor concerns have been sufficiently addressed.

I am thus pleased to inform you that your manuscript has been accepted for publication in the EMBO Journal.

Related, I would like to hereby ask your consent on keeping the referee response figures included in this file.

On a different note, I would like to alert you that EMBO Press offers a format for a video-synopsis of work published with us, which essentially is a short, author-generated film explaining the core findings in hand drawings, and, as we believe, can be very useful to increase visibility of the work. Please see the following link for representative examples and their integration into the article web page:

<https://www.embopress.org/doi/full/10.1038/s44318-025-00417-0>

Finally, we have noted that the submitted version of your article is also posted on the preprint platform bioRxiv. We would appreciate if you could alert bioRxiv on the acceptance of this manuscript at The EMBO Journal in order to allow for an update of the entry status. Thank you in advance!

Best regards,

Daniel

Daniel Klimmeck, PhD
Senior Editor
The EMBO Journal
EMBO

Postfach 1022-40
Meyerhofstrasse 1
D-69117 Heidelberg
contact@embojournal.org
